# LPCAT1 controls phosphate homeostasis in a zinc-dependent manner

Mushtak Kisko[1], Nadia Bouain[1], Alaeddine Safi[1], Anna Medici[1], Robert C Akkers[2], David Secco[1], Gilles Fouret[3], Gabriel Krouk[1], Mark GM Aarts[2], Wolfgang Busch[4,5], Hatem Rouached[1†]*

[1]BPMP, Univ Montpellier, CNRS, INRA, SupAgro, Montpellier, France; [2]Laboratory of Genetics, Wageningen University, Wageningen, Netherlands; [3]Unité Mixte de Recherche, Montpellier, France; [4]Gregor Mendel Institute, Austrian Academy of Sciences, Vienna Biocenter, Vienna, Austria; [5]Plant Molecular and Cellular Biology Laboratory, Salk Institute for Biological Studies, La Jolla, United States

*For correspondence:
hatem.rouached@supagro.inra.fr

Present address: [†]Department of Plant Biology, Carnegie Institution for Science, Stanford, United States

Competing interests: The authors declare that no competing interests exist.

**Abstract** All living organisms require a variety of essential elements for their basic biological functions. While the homeostasis of nutrients is highly intertwined, the molecular and genetic mechanisms of these dependencies remain poorly understood. Here, we report a discovery of a molecular pathway that controls phosphate (Pi) accumulation in plants under Zn deficiency. Using genome-wide association studies, we first identified allelic variation of the *Lyso-PhosphatidylCholine (PC) AcylTransferase 1* (*LPCAT1*) gene as the key determinant of shoot Pi accumulation under Zn deficiency. We then show that regulatory variation at the LPCAT1 locus contributes significantly to this natural variation and we further demonstrate that the regulation of *LPCAT1* expression involves bZIP23 TF, for which we identified a new binding site sequence. Finally, we show that in Zn deficient conditions loss of function of *LPCAT1* increases the phospholipid Lyso-PhosphatidylCholine/PhosphatidylCholine ratio, the expression of the Pi transporter PHT1;1, and that this leads to shoot Pi accumulation.
DOI: https://doi.org/10.7554/eLife.32077.001

## Introduction

All living organisms require an adequate supply of nutrients for growth and survival. Nutrient deficiencies lead to decreased plant survival and lower nutritional value of foods, which have a profound impact on human health (*Myers et al., 2014*). In particular, zinc (Zn) and iron (Fe) deficiencies affect up to 2 billion people worldwide (*Hilty et al., 2010*). According to the World Health Organization, about 800,000 child deaths per year are attributable to Zn deficiency alone (*Akhtar, 2013*). The widespread occurrence of deficiencies in micronutrients such as Zn and Fe in human populations is due to low dietary intake (*Rouached, 2013*; *Myers et al., 2014*; *Shahzad et al., 2014*). In the light of crop optimization for yield and nutritional quality, it is therefore an important goal to understand the genetic and molecular basis of plant nutrition. A complicating circumstance is that plant uptake, storage and use of these nutrients are partly dependent of each other (*Rouached and Rhee, 2017*). For instance, physiological Zn deficiency leads to over-accumulation of phosphorus (P) in the shoots (for review, *Bouain et al., 2014*; *Kisko et al., 2015*). Noteworthy, when the Zn supply is low, increasing P supply causes a reduction of plant height, delayed development and severe leaf symptoms including chlorosis and necrosis (*Ova et al., 2015*). At high P supplies, Zn deficiency associated with elevated shoot P levels causes P toxicity (*Marschner, 2012*). Interestingly, this P-Zn interaction is also recognized in a wide variety of other biological systems, including rats (*Wallwork et al., 1983*), human cells (*Sandström and Lönnerdal, 1989*), and multiple fungal species (*Freimoser et al., 2006*). In *Saccharomyces cerevisiae* yeast, the Zn status acts as a major determinant of the ability to

store P (*Simm et al., 2007*). Much like Zn nutrition, P homeostasis is of global relevance as current agricultural practices require large amounts of P. At the same time, world-wide P reserves are becoming increasingly scarce and a potential P crisis looms for agriculture at the end of this 21 st century (*Abelson, 1999*; *Neset and Cordell, 2012*). How P and Zn homeostases are coordinated is therefore not only a fundamental biological question but has also serious implications for global agronomic and biotechnological applications.

P is a critical component of many metabolites and macromolecules, including nucleic acids and phospholipids (PLs) (*Poirier and Bucher, 2002*; *Rouached et al., 2010*). Of equal importance, Zn provides chemical, structural and regulatory functions in biological systems (*Christianson, 1991*), for instance as cofactor for hundreds of enzymes, or by binding to PLs to maintain membrane structure (*Binder et al., 2001*; *Sinclair and Krämer, 2012*). Plants have evolved the ability to adjust to large fluctuations in external P or Zn supply. P is taken up by the root system in the form of inorganic phosphate (Pi). In *Arabidopsis thaliana* (Arabidopsis), this uptake relies on members of the high affinity Pi transporter family (PHT1) (*Nussaume et al., 2011*), of which PHT1;1 is the major contributor (*Ayadi et al., 2015*). Upon P deficiency, the expression of some *PHT1* transporters increases as a result of the activation of the 'PHR1-miR399-PHO2' signalling pathway (*Bari et al., 2006*; *Lin et al., 2008*; *Pant et al., 2008*), causing a strong increase in the acquisition of Pi and its subsequent translocation to the shoots (*Lin et al., 2008*; *Pant et al., 2008*). In contrast to our understanding of the molecular mechanisms involved in sensing and signalling of Pi abundance (*Chiou and Lin, 2011*; *Zhang et al., 2014*), little is known about how plants sense and signal Zn deficiency. A putative working model of Zn deficiency signalling was proposed by *Assunção et al. (2013)*, which is centred around two essential members of the bZIP transcription factor (TF) family in Arabidopsis, bZIP19 and bZIP23, without which plants are unable to respond to Zn starvation by inducing the expression of genes involved in Zn uptake and distribution such as the zinc transporter ZIP4 (*Assunção et al., 2010*). Beyond common set of genes targeted by these two TFs, each TF could regulate distinct genes (*Inaba et al., 2015*), but the identity of distinctive binding site recognized by each remains poorly unknown. Identifying such binding motif is necessary to better understand how plants regulate Zn homeostasis.

The interaction between Zn and Pi homeostasis in plants is also obvious at the molecular level (for reviews, *Bouain et al., 2014*; *Kisko et al., 2015*). For instance, Zn deprivation causes an up-regulation of *PHT1;1* and consequently an over-accumulation of Pi in *Arabidopsis thaliana* (*Jain et al., 2013*; *Khan et al., 2014*). The expression of Pi uptake transporters is normally tightly controlled in roots in response to the P status of the plant, but it is clear that this tight control is lost under Zn deficiency. Remarkably, although the involvement of PHOSPHATE RESPONSE1 transcription factor (PHR1) in the coordination of Pi-Zn homeostasis has been demonstrated (*Khan et al., 2014*), the Zn deficiency-induced Pi uptake transporter expression is independent of the aforementioned canonical 'PHR1-miR399-PHO2' signalling pathway (*Khan et al., 2014*), indicative of room for new discoveries in Pi homeostasis under Zn deficiency in plants.

In this study, we set out to identify the genes controlling such novel mechanisms to cause Pi accumulation in shoots of Zn-deficient Arabidopsis plants. Genome wide association (GWA) mapping was employed using a subset of 223 Arabidopsis accessions from the RegMap panel (*Horton et al., 2012*), which enabled us to demonstrate that there is heritable natural variation of Pi accumulation in responses to Zn deficiency and that one major locus governing this is the *LysoPhosphatidylCholine AcylTransferase 1* (*LPCAT1*) gene. Under Zn deficiency, *lpcat1* mutants showed an alteration in the phospholipids *Lyso-PhosphatidylCholine/PhosphatidylCholine* (Lyso-PC/PC) ratio, and an up-regulation of the expression of the main high affinity Pi transporter gene *PHT1;1*. Finally, we demonstrate that *LPCAT1* acts downstream of one of the two key Zn starvation signalling TFs, bZIP23, for which we identified a new binding site sequence. Overall, this study uncovered a novel pathway, in which *LPCAT1* plays a key role in the coordination of Pi homeostasis and Zn deficiency response in plants through modulation of phospholipid metabolism and Pi transporter expression.

## Results

### GWAS identify two candidate genes involved in the accumulation of Pi in the shoot under zn deficiency

To identify genes regulating shoot Pi concentration under Zn deficiency, genome wide association studies (GWAS) were conducted. To do so, a diverse set of 223 Arabidopsis accessions, selected from the RegMap panel (*Horton et al., 2012*) was grown on agar medium supplemented with (+Zn) or without Zn (–Zn) for 18 days, before assessing their shoot Pi concentration (*Supplementary file 1*). As expected, Zn deficiency in shoots of Col-0 plants was associated with the induction of the expression of two Zn-deficiency marker genes, *ZIP4* and *ZIP12* (*Jain et al., 2013*) (*Figure 1—figure supplement 1*). Under the +Zn condition, shoot Pi concentration varied across the 223 accessions from 3 to 10 µmol of Pi per gram of fresh weight (median ~5.45 µmol.gram$^{-1}$ fresh weight of Pi) (*Figure 1—figure supplement 2A*) while in -Zn, it increased to 4–16 µmol of Pi per gram of fresh weight (median ~8.23 µmol.gram$^{-1}$ fresh weight of Pi) (*Figure 1—figure supplement 2B*). .The broad-sense heritability (H$^2$) of the shoot Pi concentrations was 0.63 and 0.47 under +Zn and –Zn conditions, respectively. Using the genotype and the shoot Pi concentration as input, we performed a mixed model (AMM method [*Seren et al., 2012*]) GWAS that corrects for population structure (*Korte et al., 2012*) for both Zn conditions (*Figure 1B,C*, *Figure 1—figure supplement 2C–D*). Using a non-conservative 10% false discovery threshold (FDR), we identified 13 significant SNPs in five distinct genomic loci to be associated with Pi concentration in the shoots, which was specific for the –Zn condition (*Figure 1B*). The most significantly associated SNP (p-value=5.86*10$^{-8}$; FDR 1%) was located on Chromosome 1 (*Supplementary file 2*). A haplotype analysis centred on the 50 kb region surrounding the significantly associated SNP revealed one main haplotype (depicted in purple) that was associated with the marker SNP and high Pi concentration (*Figure 1—figure supplement 3*). The significantly associated SNP was located at the upstream and coding regions of two candidate genes, namely *At1g12640* and *At1g12650* (*Figure 1D–E*). *At1g12650* encodes an unknown protein likely to be involved in mRNA splicing via the spliceosome, and *At1g12640* encodes a member of the *Membrane Bound O-Acyl Transferase* (*MBOAT*) gene family known as *Lyso-PhosphatidylCholine AcylTransferase 1* (*LPCAT1*, [*Wang et al., 2012*]). LPCAT1 is an evolutionarily conserved key enzyme that is involved in phospholipid metabolism and more precisely in the Lands cycle (*Lands, 1960*). In Arabidopsis LPCAT1 has been shown to catalyze the conversion of lysophosphatidylcholine (Lyso-PC) to produce phosphatidylcholine (PC) (*Zheng et al., 2012*).

### *LPCAT1* is involved in regulating shoot Pi concentration in Zn deficiency

In order to determine the causal gene underlying the shoot Pi accumulation Quantitative Trait Locus (QTL) in –Zn, we used a reverse genetic approach. The first thing we studied was to test if any of these two genes is indeed involved in the -Zn-specific variation in shoot Pi concentration. Therefore, wild-type Arabidopsis (Columbia-0, Col-0), T-DNA insertion mutant lines for *LPCAT1* (At1g12640) (*Wang et al., 2012*) and for *At1g12650* gene were grown for 18 days on +Zn or –Zn media before assessing their shoot Pi concentration. In response to -Zn, Col-0 plants showed a significant increase (~29% increase, p-value<0.05) in their shoot Pi concentration compared to +Zn conditions (*Figure 2A*), which is in line with a previous report (*Khan et al., 2014*). Importantly, while Pi accumulation in response to –Zn in *At1g12650* mutants was indistinguishable from Col-0, *lpcat1* mutants displayed a significant increase in shoot Pi concentration (~36% increase, p-value<0.05) (*Figure 2A*). We confirmed that this increase in shoot Pi concentration in the *lpcat1* mutants is specific to the -Zn treatment as no significant differences were observed in the +Zn condition compared to Col-0. These results showed that *LPCAT1*, and not *At1g12650*, is involved in regulating shoot Pi concentration in response to Zn deficiency in Arabidopsis. Our further efforts were therefore directed at understanding the transcriptional regulation of *LPCAT1* by –Zn, and then at resolving how allelic variation at the *LPCAT1* gene contributes to the variation in shoot Pi concentration.

### *LPCAT1* acts downstream of bZIP23 transcription factor

To investigate the molecular causational links between Zn/LPCAT1/Pi, we analyzed the *cis*-regulatory elements present within the 1500 bp region upstream of the *LPCAT1* start codon (in Col-0 background) using the search tool AthaMap (*Bülow et al., 2010*). We identified the presence of a single

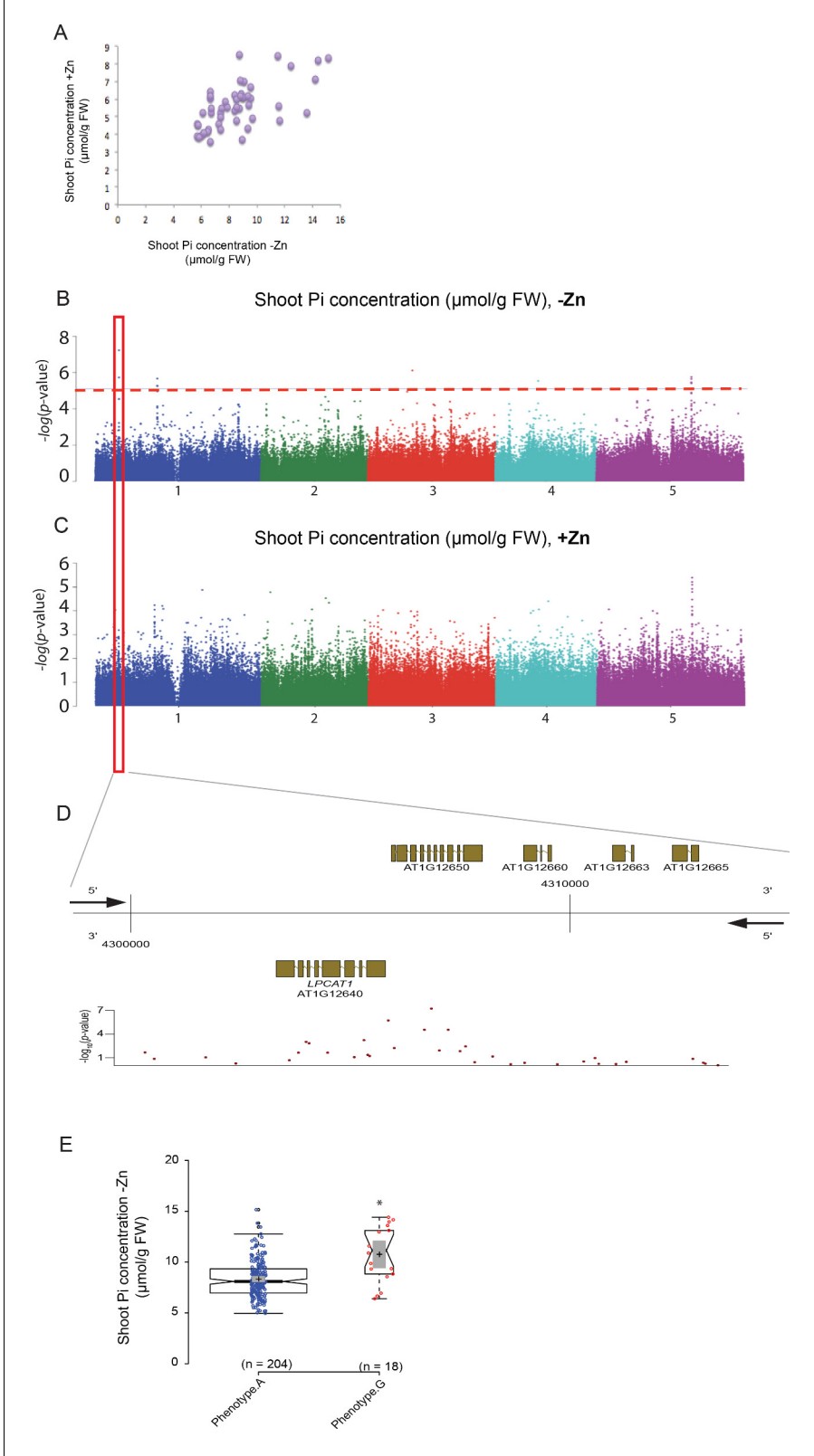

**Figure 1.** Genome-wide association (GWA) analysis of Arabidopsis shoot Pi concentration. 223 Arabidopsis thaliana accessions were grown supplemented with zinc (+Zn) or without zinc (-Zn) for 18 days under long day conditions, upon which shoot inorganic phosphate (Pi) concentrations were determined. (A) Mean shoot Pi concentration of Arabidopsis accessions in +Zn is plotted against the mean shoot Pi concentration of Arabidopsis accessions in -Zn. (B, C) Manhattan plots of GWA analysis of Arabidopsis shoot Pi concentration in -Zn (B) and +Zn (C). The five Arabidopsis

*Figure 1 continued on next page*

*Figure 1 continued*

chromosomes are indicated in different colours. Each dot represents the –log10(P) association score of one single nucleotide polymorphism (SNP). The dashed red line denotes an approximate false discovery rate 10% threshold. Boxes indicate the location of the *LPCAT1* (red) quantitative trait loci (QTL). (D) Gene models (upper panel) and SNP –log10(P) scores (lower panel) in the genomic region surrounding the GWA QTL at the *LPCAT1* (E); 5' and 3' indicate the different genomic DNA strands and orientation of the respective gene models. (E) Distribution of Pi concentrations in accessions with the phenotype A versus accessions with phenotype G. Asterisk indicates a significant difference between the two groups of accessions of p<0.01.

DOI: https://doi.org/10.7554/eLife.32077.002

The following figure supplements are available for figure 1:

**Figure supplement 1.** mRNA abundance of Zn-responsive genes *ZIP4* and *ZIP12* in roots of Col-0 plants grown in presence and absence of Zn.

DOI: https://doi.org/10.7554/eLife.32077.003

**Figure supplement 2.** Genome-wide association (GWA) analysis of Arabidopsis shoot Pi concentration.

DOI: https://doi.org/10.7554/eLife.32077.004

**Figure supplement 3.** Haplotype analysis of region around SNP C1P4306845.

DOI: https://doi.org/10.7554/eLife.32077.005

copy of the 10 bp Zinc Deficiency Response Element (ZDRE, RTGTCGACAY)(*Assunção et al., 2010*), located 377 bp upstream of the ATG (*Figure 2B*). This motif is a known binding site for the bZIP19 and bZIP23 transcription factors, the key transcriptional regulators of the –Zn response (*Assunção et al., 2010*). Given the presence of the ZDRE, we hypothesized that the expression of *LPCAT1* under –Zn could be controlled by the bZIP19 or bZIP23 TFs. An electrophoretic mobility shift assay (EMSA) was performed, using a 30 bp promoter fragment containing the 10 bp potential ZDRE, which confirmed that both bZIP19 and bZIP23 could bind to this *cis*-regulatory element (*Figure 2C*), as had already been shown by (*Assunção et al., 2010*).

Further analysis of the regulatory regions of *LPCAT1* led us to identify a new motif GTGTCGAA (5' untranslated region of *LPCAT1*), very similar to that of the ZDRE motif (RTGTCGACAY) (*Figure 2B*). Due to the sequence similarity of this newly identified motif to that of ZDRE, we first tested the capacity of bZIP23 or bZIP19 to bind to this motifs. Interestingly, EMSA analysis revealed that bZIP23 could bind to the newly identified motif, while bZIP19 showed an extremely weak (if any) binding capacity to new motif (*Figure 2C*). These findings strongly support the Zn-dependency of *LPCAT1* expression. We therefore determined the transcript abundance of *LPCAT1* in shoots of Arabidopsis wild-type plants (Col-0) grown in -Zn for 6, 12 and 18 days. In response to -Zn, transcript accumulation of *LPCAT1* was changed, showing significant down-regulation compared +Zn conditions (*Figure 2D*). This result shows that repression of *LPCAT1* upon low –Zn is associated with higher Pi levels and suggests that transcriptional regulation of *LPCAT1* is important for its involvement in Pi homeostasis. We next tested whether these bZIP TFs could be involved in regulating the expression of *LPCAT1* in –Zn. To test this, we determined the expression levels of *LPCAT1* in the *bzip19* and *bzip23* single and *bzip19/bzip23* double knock-out mutant lines and WT plants (Col-0) grown for 18 days in +Zn and –Zn conditions. The *LPCAT1* transcript was significantly up-regulated in the *bzip23* and *bzip19/bzip23* mutant lines, compared to Col-0 and *bzip19* in –Zn, which showed a significant down-regulation (*Figure 2D*). This indicates that bZIP23, but not bZIP19, is involved in negatively regulating the expression of *LPCAT1* under –Zn. We therefore hypothesized that *bZIP23* but not bZIP19 are necessary for the downregulation of *LPCAT1* in –Zn and subsequent Pi accumulation and there assessed the capacity of the mutants to accumulate Pi when grown with or without Zn for 18 days. While in +Zn, all plants showed similar shoot Pi content, we observed a significant decrease in shoot Pi content in the *bzip23* and *bzip19/bzip23* mutants compared to Col-0, confirming the regulatory role of bZIP23 and not bZIP19 (*Figure 2E*). Taken together, this suggests that bZIP23 represses *LPCAT1* upon –Zn, and this repression leads to the over-accumulation of Pi in shoots in Arabidopsis grown under –Zn condition.

## Allelic variation of *LPCAT1* determines natural variation of Pi content under zinc deficiency

We next wanted to test whether allelic variation of *LPCAT1* is causal for the observed differences in Pi accumulation under –Zn. For this, we selected two contrasting groups of accessions with either a high ratio (Br-0, Ts-1, PHW-2 and Sap-0) or a low ratio (Ang-0, CIBC-5, Col-0, EST-1, RRS-10) of Pi accumulated in shoots of -Zn plants compared to +Zn plants (*Figure 3A*). Interestingly, comparative

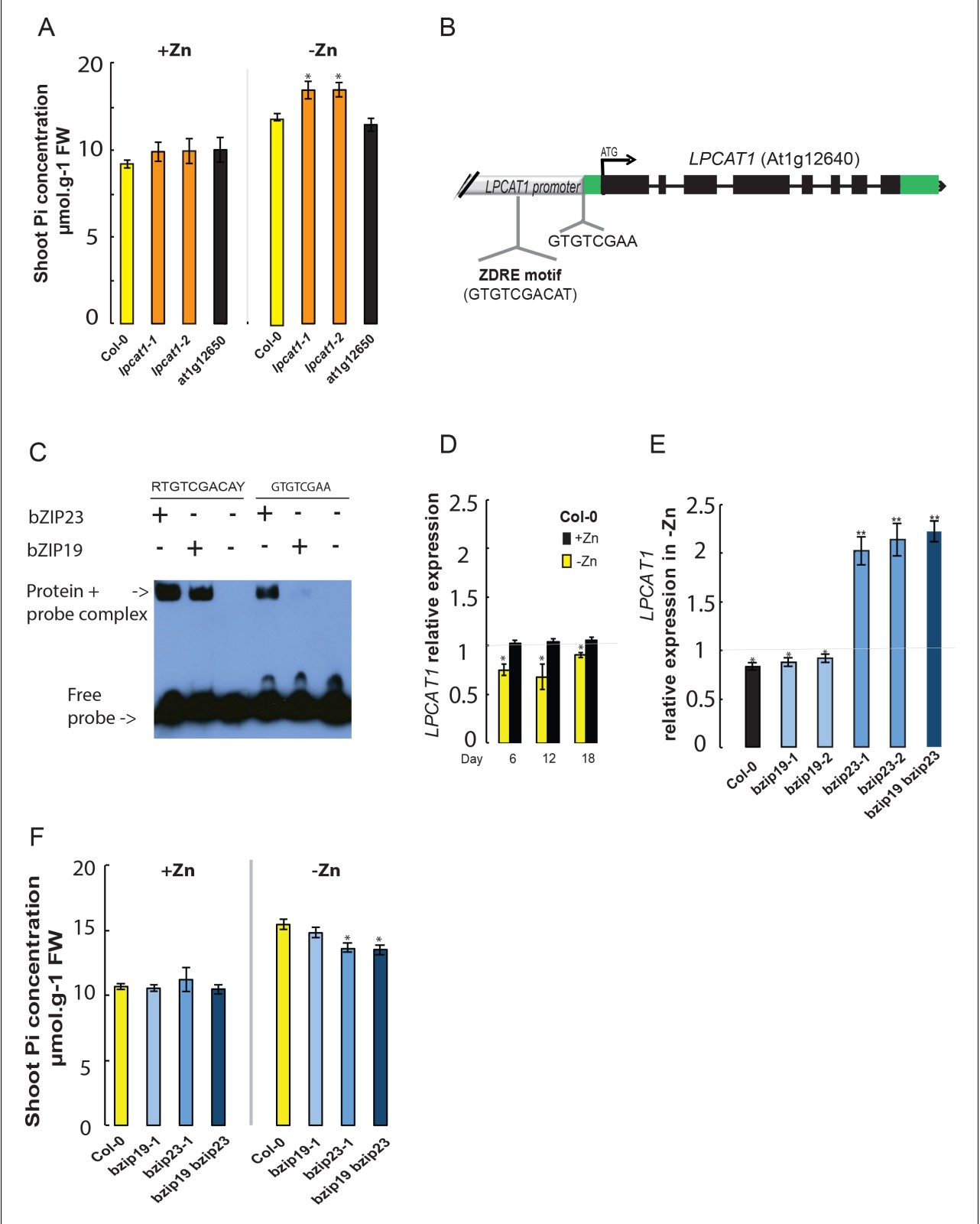

**Figure 2.** Loss of function mutation of *Lyso-PhosphatidylCholine AcylTransferase 1* (*LPCAT1*), and not At1g12650, affects shoot Pi concentration in a Zn supply and bZIP23 dependent manner. (**A**) Shoot Pi concentration of 18-days-old Col-0 wild-type plants, lpcat1 and At1g12650 mutants grown in +Zn or –Zn conditions. (**B**) Gene structure of *LPCAT1*. The grey box represents the promoter region, green boxes are 5' and 3' untranslated regions, black boxes represent exons, and black lines represent introns, the arrow head indicates the direction of transcription, ATG indicates the start codon. The

*Figure 2 continued on next page*

*Figure 2 continued*

Zinc Deficiency Response Element (ZDRE) binding site for bZIP19 and bZIP23, and the newly identified binding site for bZIP23 are indicated. (C) Differential binding of bZIP19 and bZIP23 to two promoter regions of LPCAT1 gene. EMSA analysis on 30 bp promoter fragments from motif present in *LPCAT1* promoter of contrasting accessions showed in (A). (D) Relative *LPCAT1* transcript abundance (-Zn/+Zn) in Col-0 wild-type plants grown on +Zn or -Zn agar medium for 6, 12 and 18 days. (E) Relative *LPCAT1* transcript abundance in Col-0 wild-type plants, *bzip19*, *bzip23*, and *bzip19/bzip23* double mutants grown on +Zn or -Zn agar medium for 18 days. The relative mRNA levels was quantified by RT-qPCR and normalized to the *Ubiquitin10* reference mRNA level (*UBQ10*: At4g05320). (F) Shoot Pi concentration in Col-0 wild-type plants, *bzip19* and *bzip19/bzip23* double mutants grown on +Zn or -Zn agar medium for 18 days. Values are means of three to six biological replicates. Individual measurements were obtained from the analysis of shoots collected from a pool of 10 plants. Error bars indicate SD; One and two asterisks indicate a significant difference with WT plants (ANOVA and Tukey test) of p<0.05 and p<0.01, respectively.

DOI: https://doi.org/10.7554/eLife.32077.006

sequence analysis of the regulatory regions of *LPCAT1* of these accessions using the sequence data from the 1001 genomes project (*1001 Genomes Consortium, 2016*) revealed that the common ZDRE motif (*Figure 2B*) didn't display any variation between these two groups of accession (*Figure 3A*), and that the newly identified *bzip23* specific motif (*Figure 2B*) showed clear variation between the two groups of accession with the accessions with low Pi ratio under -Zn exhibiting a Col-0 like GTGTC<u>G</u>AA motif and the high Pi accumulating accession displaying a GTGTC<u>AC</u>A motif (*Figure 3A*, *Figure 3—figure supplement 1*, *Supplementary file 3*). We therefore tested the capacity of bZIP23 and bZIP19 to bind to this latter version (GTGTC<u>AC</u>A) of the ZDRE motif. EMSA analysis revealed that only bZIP23 could bind to this version of ZDRE motif (*Figure 3B*). Taken together, EMSA results (*Figure 2C*, *Figure 3B*) support the specificity of a new ZDRE motif for bZIP23.

We next assessed the effect of these motif sequence changes on the activity of the *LPCAT1* promoter using a quantitative *in planta* transactivation assay (*Bossi et al., 2017*). In this assay, we co-transformed tobacco leaves with an effector construct (35S::bZIP23 or 35S::YFP) with a reporter construct, containing either the LPCAT1 Col-0 native promoter (with GTGTC<u>G</u>AA motif), the LPCAT1 Col-0 mutated promoter (with GTGTC<u>AC</u>A motif), or the promoter of the zinc transporter *ZIP4* promoter (as positive control) fused to a *β-glucuronidase* (*GUS*)-encoding reporter gene (*Figure 3C*). The comparison of the ability of 35S-bZIP23 or 35S-YFP to activate these *LPCAT1* promoters was performed by quantifying the GUS activity. The average relative activity was calculated as the GUS activity of each promoter in the presence of 35S::bZIP23 divided by its GUS activity in the presence of 35S::C-YFP (*Figure 3D*). As expected, our results showed an induction of the positive control (*ZIP4* promoter by 35S:bZIP23). Consistent with the hypothesis that the natural variation of the new ZDRE is relevant for *LPCAT1* regulation, bZIP23 acted as stronger repressor of the *LPCAT1* Col-0 mutated promoter that contained the new ZDRE of Sap-0 compared to the Col-0 *LPACT1* native promoter (*Figure 3D*). Moreover, *LPCAT1* was downregulated by a larger extent in accessions that accumulated more Pi upon –Zn (*Figure 3E*) and contained the new (non Col-0) ZDRE while the expression of *bZIP23* remained unchanged in all accessions and growth conditions tested.

To further test whether the difference in *LPCAT1* expression was due to the natural allelic variation in the regulatory regions and whether this was causal for the Pi accumulation, we focused on only two contrasting accessions, Sap-0 and Col-0, which displayed a significantly different capacity to accumulate shoot Pi in –Zn (*Supplementary file 1*). Noteworthy, the *LPCAT1* promoter and predicted amino acid coding sequences of Col-0 and Sap-0 displayed 97.9% and 99.4% sequence identity respectively (data not shown). The *lpcat1* knock-out mutant (in Col-0 background) was then transformed with either an empty vector (control) or one of four constructs containing 1.5 kbp of the promoter (immediately upstream of the start codons) of either pLPCAT1[Col-0] or pLPACT1[Sap-0] respectively fused to either the coding region of *LPCAT1*[Col-0] or *LPCAT1*[Sap-0] (*Figure 4A*). Three independent, single locus insertion lines (based on segregation of the insertion in progeny of a hemizygous plant) were considered for the analysis. When expressed under the pLPCAT1[Col-0] promoter, *LPCAT1*[Col-0] or *LPCAT1*[Sap-0] complemented the *lpcat1-1* knock-out mutant phenotype and showed a similar Pi content to WT (Col-0) plants in both +Zn and –Zn conditions (*Figure 4B*). This indicates that the polymorphisms in the coding region are not responsible for the change in Pi content in –Zn conditions. In contrast, lines complemented with the pLPCAT1[Sap-0]:LPCAT1[Col-0] or pLPCAT1[Sap-0]:LPCAT1[Sap-0] transgenic lines showed significantly higher Pi content compared to pLPCAT1[Col-0]:LPCAT1[Col-0] or pLPCAT1[Col-0]:LPCAT1[Sap-0] lines or WT (Col-0) in -Zn conditions (*Figure 4B*). This

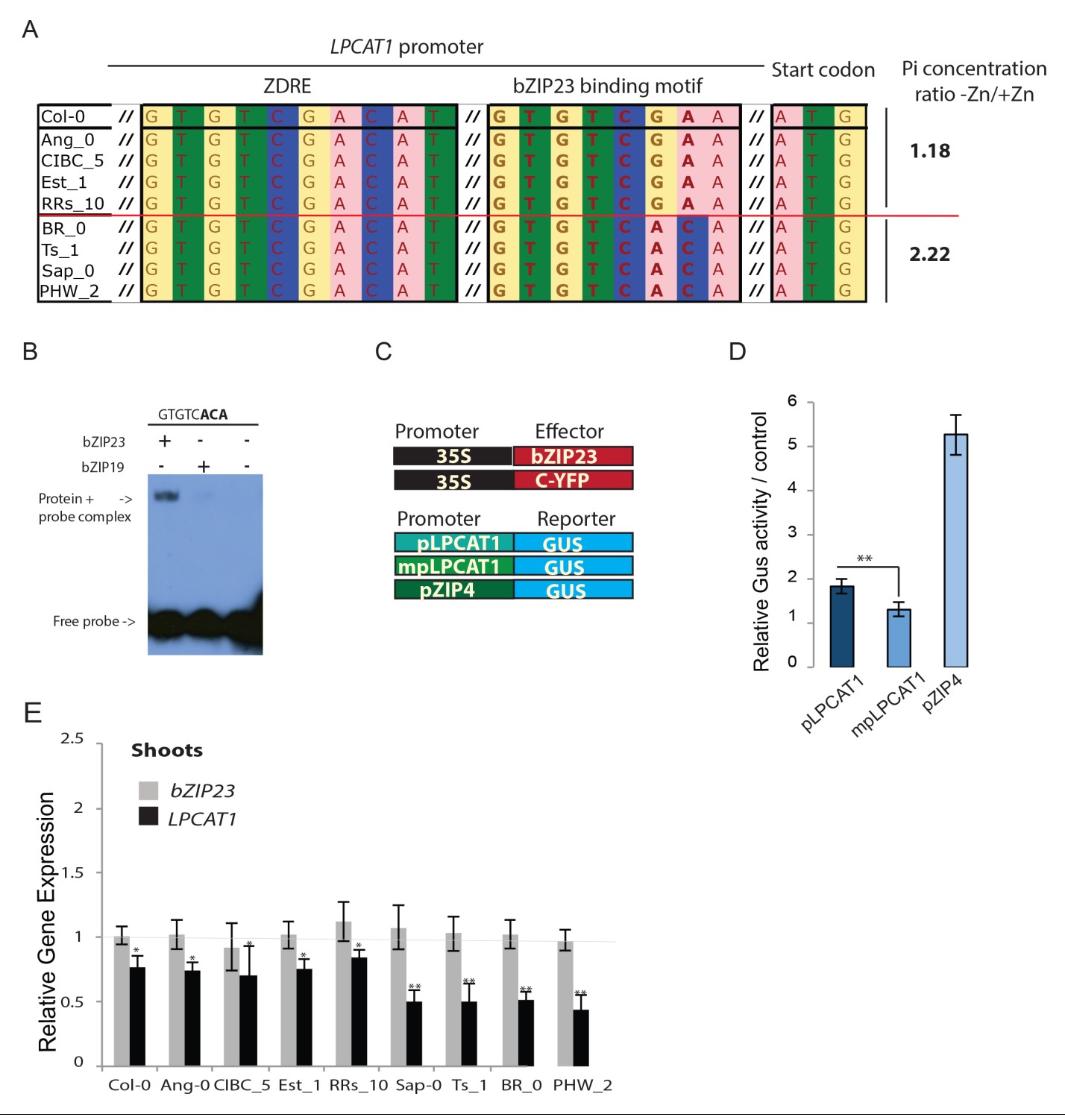

**Figure 3.** Identification of a new binding motif specific for bZIP23, and the variation of LPCAT1 gene expression between genotypes in -Zn condition. (**A**) Sequence comparison for ZDRE and the new binding site motif for bZIP23 in the promoter of accession with high ratio of Pi accumulation in -Zn/+Zn (Col-0, Ang-0, CICB-5, Est-1, RRS-10) and low Pi accumulation ratio - Zn/+Zn (Sap-0, Ts-1, Br-0 and PHW-2). (**B**) Differential binding of bZIP19 and bZIP23 to a specific ZDRE motif of *LPCAT1* (GTGTCACA). (**C–D**) *in planta* transactivation assay. (**C**) 35S:bZIP23 and 35S::C-YFP were used as effectors. p*LPCAT1*: the native Col-0 *LPACT1* promoter (with « GTGTCGAA » as new ZDRE), mp*LPCAT1*: a modified (point mutation) version of the Col-0 promoter to only contain the new ZDRE of Sap-0 (« GTGTCACA »); pZIP4; promoter of the zinc transporter *ZIP4* gene. Each p*LPCAT1* (native, Col-0), mp*LPCAT1* (mutated version) or p*ZIP4* promoter was fused to a *β-glucuronidase (GUS)*-encoding reporter gene (reporter). (**D**) The effect of bZIP23 TF on the activity of each promoter p*LPCAT1*, mp*LPCAT1* or p*ZIP4* was determined by measuring GUS activity. The effect of C-YFP protein on the activity

*Figure 3 continued on next page*

Figure 3 continued

of each promoter p*LPCAT1*, mp*LPCAT1* or p*ZIP4* was used as a control to determine the basal level of GUS activity for each promoter. Comparing the effect of bZIP23 TF and C-YFP protein on each promoter enabled the determination of the relative GUS activity. Error bars represent standard error from three independent experiments. The asterisks indicate that the relative GUS activity is statistically different from the YFP control (p-value<0.01, t-test). (E) Relative *bZIP23* and *LPCAT1* transcripts abundance in -Zn and +Zn conditions. Col-0, Ang-0, CICB-5, Est-1, RRS-10, Sap-0, Ts-1, Br-0 and PHW-2 genotypes were grown on +Zn or -Zn agar medium. The relative mRNA level was quantified by RT-qPCR and normalized to the *Ubiquitin10* reference mRNA level (*UBQ10*: At4g05320). Values are means of six biological replicates. Individual measurements were obtained from the analysis of shoots collected from a pool of 20 plants. Error bars indicate SD; one and two asterisks indicate a significant difference with Col-0 plants (ANOVA and Tukey test) of p<0.05 and p<0.01 respectively.
DOI: https://doi.org/10.7554/eLife.32077.007

The following figure supplement is available for figure 3:

**Figure supplement 1.** Shoot Pi concentrations in Arabidopsis accessions grouped by new ZDRE motif.
DOI: https://doi.org/10.7554/eLife.32077.008

result demonstrates that regulatory variation in of the LPCAT1 promotor determines Pi accumulation and favours the model that variation in the expression level of *LPCAT1* as the cause of the variation in Pi accumulation in –Zn. Therefore, we assessed *LPCAT1* mRNA accumulation in in WT (Col-0) and all transgenic lines grown in both +Zn and –Zn conditions. Our result showed that while *LPCAT1* is down-regulated in all tested lines by –Zn treatments, the lines complemented with the *LPCAT1* driven by p*LPCAT1* [Sap-0] accumulates significantly lower *LPCAT1* mRNA than that of those under the control of p*LPCAT1* [Col-0] and WT (Col-0) (**Figure 4C**). Taken together, our results indicate that the allelic variation between Col-0 and Sap-0 in the promoter of the *LPCAT1* gene causes the difference in *LPCAT1* expression, and confirm that this difference leads to the difference in Pi accumulation under –Zn. Importantly, the polymorphisms in the *bzip23* binding site in the promotor of *LPCAT1* suggest a potential cis-regulatory mechanism for this.

## *LPCAT1* mutation impacts phospholipid concentrations in –Zn

While *LPCAT1* had not been implicated in any known process involving Zn, it is known to catalyse the conversion of lyso-phosphatidylcholine (Lyso-PC) to phosphatidylcholine (PC) in the remodelling pathway of PC biosynthesis (**Figure 5A**) (*Lands, 1960*; *Chen et al., 2007*; *Wang et al., 2012*). Consequently, we hypothesized that a mutation in *LPCAT1* or *bZIP23* would affect the Lyso-PC and PC under –Zn conditions. To test this, we measured the composition of these two phospholipid classes in the shoots of the Col-0 wild type and the *bzip23* and *lpcat1* mutants, in +Zn and –Zn conditions. In +Zn, no significant changes in the Lyso-PC and PC levels in the three different genotypes were observed (**Figure 5B,C**). However, under –Zn, *bzip23* showed a modest (but non-significant) decrease in the Lyso-PC/PC ratio while the mutation in *LPCAT1* resulted in a significant increase of Lyso-PC and a decrease of PC, resulting in an increase of the Lyso-PC/PC ratio (~1.2 fold, p-value<0.05) compared to Col-0 plants (**Figure 5D**). These results demonstrate that the LPCAT1 function is required to maintain the shoot Lyso-PC/PC ratio under –Zn. We next tested whether the polymorphisms in the regulatory region of *LPCAT1* are responsible for the change in LPC/PC ratio that ultimately affects the Pi content in –Zn conditions. We determined the LPC, PC concentrations in the shoots of the plants expressing *LPCAT1* driven by the LPCAT1[Col-0] promoter (pLPCAT1[Col-0]::*LPCAT1*[Col-0], pLPCAT1[Col-0]::*LPCAT1*[Sap-0]) or the LPCAT1[Sap-0] promoter (pLPCAT1[Sap-0]::*LPCAT1*[Col-0], pLPCAT1[Sap-0]::*LPCAT1*[Sap-0]) in the *lpcat1* mutant background, grown in presence or absence of Zn for 18 days. WT plants of the Col-0 and Sap-0 accessions, and *lpcat1* transformed with the empty vector were included in this analysis. In the presence of Zn, no difference in PC or LPCA concentrations was observed between all plant lines. However, under –Zn conditions, Sap-0, pLPCAT1[Sap-0]::*LPCAT1*[Col-0], pLPCAT1[Sap-0]::*LPCAT1*[Sap-0] or *lpcat1* lines showed a significant decrease of PC and increase of LPC concentrations, leading to an increase of Lyso-PC/PC ratios (**Figure 6**). These results further support the association between the increase of LPC/PC ratios and the alterations in P content in the plant shoots under Zn deficiency (**Figure 4b**).

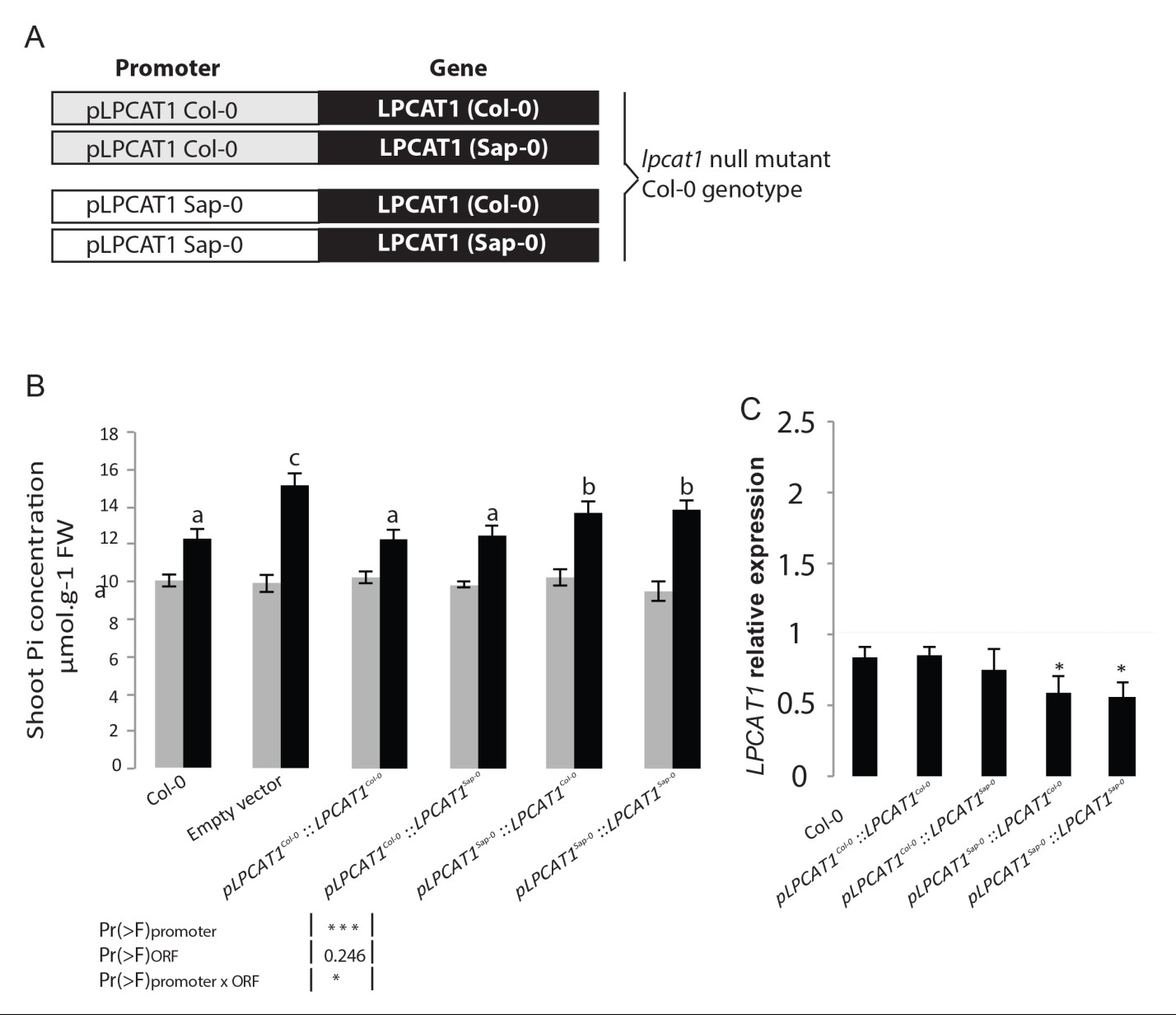

**Figure 4.** Natural allelic variation of *LPCAT1* locus causes phenotypic variation of Pi accumulation in Zn deficiency conditions. (**A**) Schematic representation of the transgenic constructs used to complement the *lpcat1* null mutant (Col-0 background). (**B**) Shoot Pi concentration (–Zn /+Zn) of 18-days-old Col-0 wild-type plants, *lpcat1* mutant transformed with empty vector, or with constructs schematized in (**A**) grown in +Zn or –Zn conditions. (**C**) The ANOVA results are presented in the table. Significative codes: '***' 0.001 and '*' 0.05. Relative *LPCAT1* transcript abundance in wild-type plants (Col-0 background) and the transgenic lines generated using the construct schematized in (**A**) grown on +Zn or -Zn agar medium. The relative mRNA levels was quantified by RT-qPCR and normalized to the *Ubiquitin10* reference mRNA level (*UBQ10*: At4g05320). Values are means of three to biological replicates. Individual measurements were obtained from the analysis of shoots collected from a pool of six plants. Error bars indicate SD; asterisks indicates a significant difference with Col-0 plants (ANOVA and Tukey test) of $p < 0.05$.

DOI: https://doi.org/10.7554/eLife.32077.009

## Accumulation of Pi in *lpcat1* involves the *HIGH AFFINITY PHOSPHATE TRANSPORTER PHT1;1*

While the molecular function of *LPCAT1* is related Lyso-PC/PC homeostasis, it doesn't answer the question how it might cause Pi levels to increase under –Zn conditions. A first hint towards answering this question came from our GWAS data: The third most significant associated peak (8% FDR)

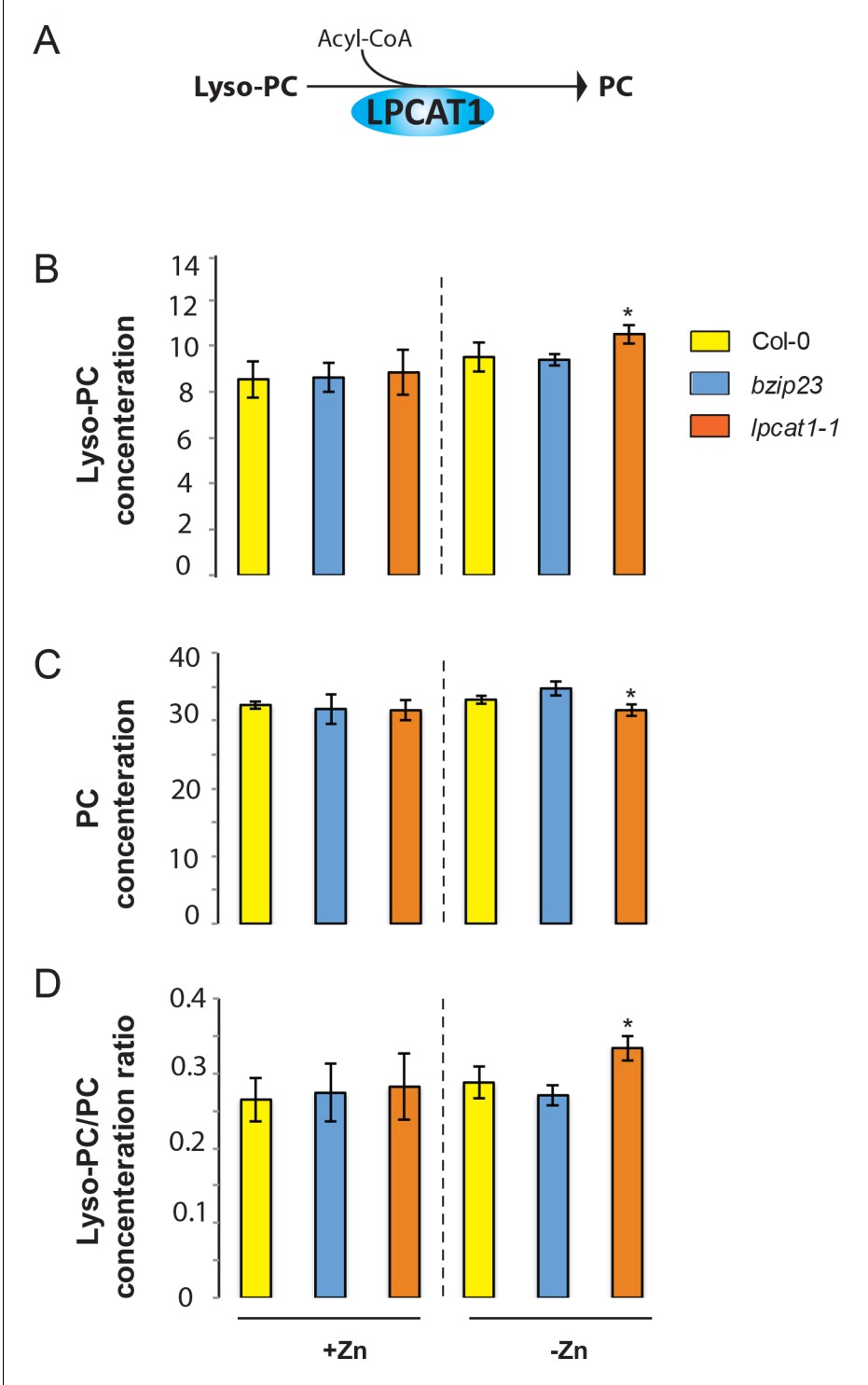

**Figure 5.** Loss of function mutations of *LPCAT1* affects the lysoPC/PC ratio in –Zn conditions. (**A**) Schematic representation of the biochemical function of LPCAT1, which catalyses the formation of phosphatidylcholine (PC) from lyso-PC and long-chain acyl-CoA. (**B**) Lyso-PC concentration (**C**) PC concentration (**D**) Lyso-PC/PC concentration ratios of Col-0 wild-type plants, *bzip23* and *lpcat1* mutant lines grown in +Zn or -Zn conditions for 18 days. Individual measurements were obtained from the analysis of shoots collected from a pool of five plants. Data are mean ±SD of three biological replicates. Statistically significant differences (ANOVA and Tukey test, p<0.05) between mutants and Col-0 are indicated with asterisks.
DOI: https://doi.org/10.7554/eLife.32077.010

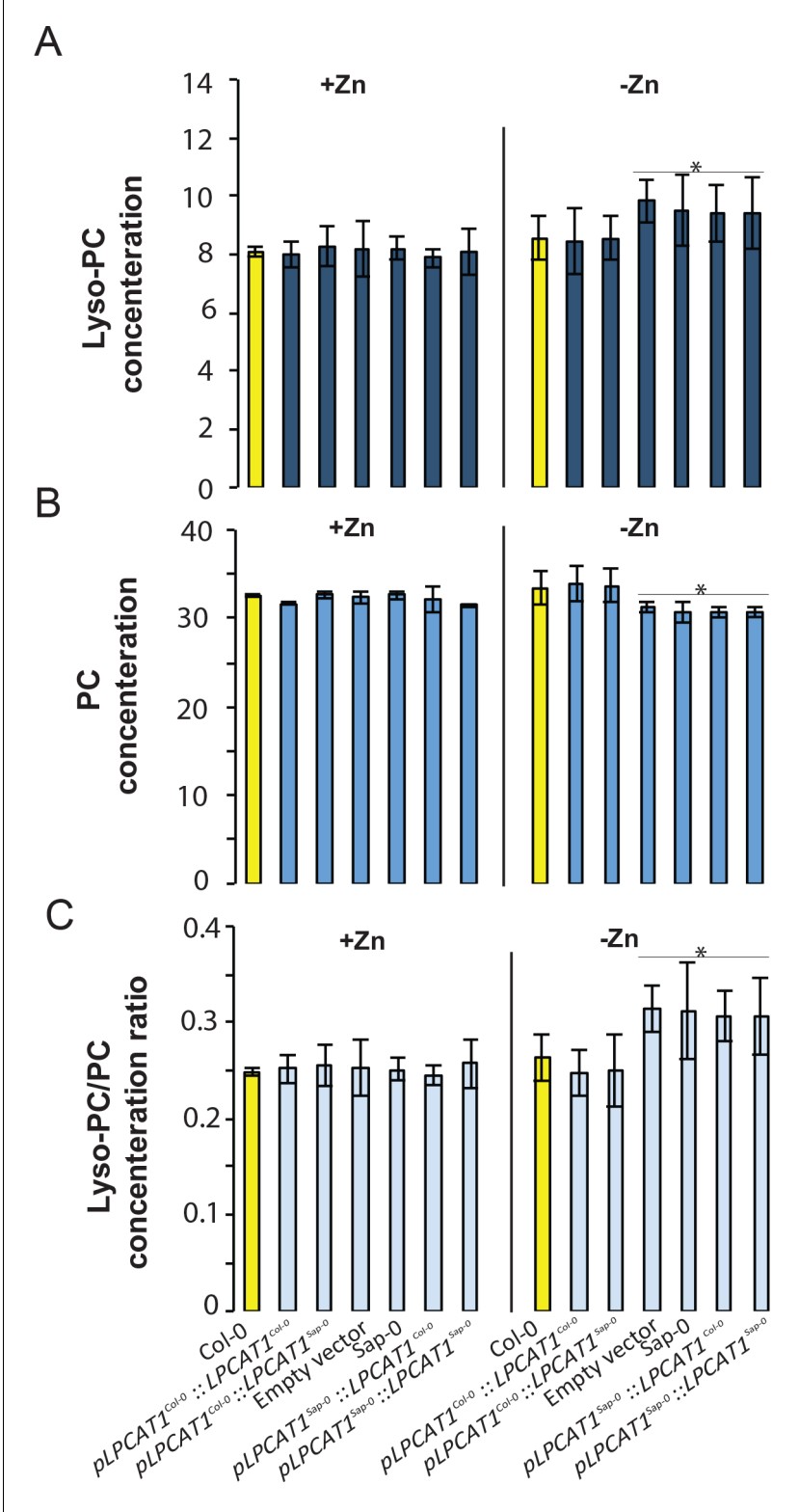

**Figure 6.** Effect of the polymorphisms in the regulatory region of *LPCAT1* on the change in LPC/PC ratios in –Zn conditions. (**A**) Lyso-PC concentration (**B**) PC concentration (**C**) Lyso-PC/PC concentration ratios of Sap-0, Col-0 wild-type plants, and *lpcat1* expressing pLPCAT1$^{Col-0}$::*LPCAT1*$^{Col-0}$, pLPCAT1$^{Col-0}$::*LPCAT1*$^{Sap-0}$, pLPCAT1$^{Sap-0}$ :: *LPCAT1*$^{Col-0}$, pLPCAT1$^{Sap-0}$::*LPCAT1*$^{Sap-0}$ constructs and *lpcat1* transformed with empty lines grown in +Zn or -Zn conditions for 18 days. Individual measurements were obtained from the analysis of shoots collected from a pool

*Figure 6 continued on next page*

*Figure 6 continued*

of five plants. Data are mean ±SD of three biological replicates. Statistically significant differences (ANOVA and Tukey test, p<0.05) between mutants and Col-0 are indicated with asterisks.

DOI: https://doi.org/10.7554/eLife.32077.011

under –Zn conditions was located in a region of chromosome 5 containing members of the high affinity Pi transporters *PHT1* gene family, namely *PHT1;1*, *PHT1;2*, *PHT1;3* and *PHT1;6* (Chr5 : 17394363–174200000) (*Figure 7A–C*, *Supplementary file 2*). Except for *PHT1;6*, the role of these genes in Pi uptake, transport and accumulation in Arabidopsis is well documented (*Nussaume et al., 2011*; *Ayadi et al., 2015*). To test the activity of one of these genes might be related to the *LPCAT1* dependent Pi accumulation under –Zn, weassessed the expression of the *PHT1* transporter genes in the shoots of *lpcat1* mutant and WT (Col-0) plants grown in +Zn or –Zn for 18 days. In all genotypes, *PHT1;1* was the only member of the *PHT1* gene family to be significantly up-regulated in the -Zn condition (*Figure 7D*). Zn deficiency induces transcription of *PHT1;1* already ~2.2 fold (p<0.05) in WT (Col-0), and this induction was further increased by 2-fold (p<0.01) in *lpcat1* mutants (*Figure 7D*), when compared to +Zn (*Figure 7—figure supplement 1*). The expression of the *PHT1;1* was thereafter tested for responsiveness to -Zn in roots of WT (Col-0) and the *lpcat1-1* mutant. While -Zn caused no significant change in expression of the *PHT1;1* in roots of WT, it increased its expression by ~2 fold in roots of *lpcat1* mutant (*Figure 7—figure supplement 2*). We next determined the effects loss of function for each phosphate transporter located under the second GWAS peak (*PHT1;1*, *PHT1;2* and *PHT1;3*) for the accumulation of Pi in -Zn in 18-day-old plants. The *pht1;1* mutant showed low Pi accumulation in presence of Zn compared to WT plants (*Figure 7E*) consistently with (*Shin et al., 2004*) that reported that the *pht1;1* mutant showed a reduction in Pi content of the shoots relative to wild type plants grown under control condition (+Pi + Zn). Importantly, no increase of Pi concentration was observed in the shoots of *pht1;1* grown in -Zn, which contrast with the Pi accumulation in *pht1;2* and *pht1;3* that was in a similar range to WT plants in presence or absence of Zn. These results show the involvement of *PHT1;1* in the over-accumulation of Pi in the shoot of *lpcat1* grown in -Zn, and further supports a second peak of the GWAS on the chromosomal region of *PHT1* genes.

## Discussion

Understanding how Zn and Pi homeostasis are wired to regulate growth is crucial to offer a new perspective of improving Pi nutrition in plants by modulating the Zn-deficiency signalling pathway. Our study provides a first insight into the genetic and molecular mechanism that controls shoot Pi concentration under –Zn in plants by discovering a pathway which includes the –Zn response TF *bZIP23* that target the *LPCAT1,* and the Pi transporter *PHT1;1*.

In *A. thaliana*, GWAS has been shown to be a powerful approach to detect loci involved in natural variation of complex traits including variation in the accumulation of non-essentials or toxic elements in plants, such as sodium (*Baxter et al., 2010*), cadmium (*Chao et al., 2012*) or arsenic (*Chao et al., 2014*). Here we used GWAS to identify genes involved in the regulation of the essential macronutrient (P) concentration in its anionic form (Pi) in plants grown under control conditions (+Zn) and -Zn. In both conditions, our GWA analysis reveals that there is widespread natural variation in shoot Pi concentration, and supports the existence of genetic factors that affect this trait (*Figure 1*). The GWAS data support the –Zn specificity of this response, since no association was detected around the *LPCAT1* locus in our control condition (+Zn) (*Figure 1*). The presence of the Zinc Deficiency Response Element (ZDRE) (*Assunção et al., 2010*) in the promoter of *LPCAT1* and more particularly the newly identified binding motif specific for bZIP23 (*Figures 2* and *3*) in the 5' untranslated leader of *LPCAT1* is a strong argument supporting the Zn-dependency of this response.

A ZDRE is present in the promoter regions of many genes targeted by bZIP19 and bZIP23 (*Assunção et al., 2010*). In addition to their positive regulatory role by inducing several Zn deficiency related genes, publicly available microarray showed that bZIP19 and bZIP23 may have a negative regulatory role as many genes were induced in the bzip19/bzip23 mutant background compared to WT plants grown in –Zn (*Azevedo et al., 2016*). A functional redundancy of these two TFs was proposed based on the oversensitivity of the *bzip19* and *bzip23* double mutant to –Zn, which was not

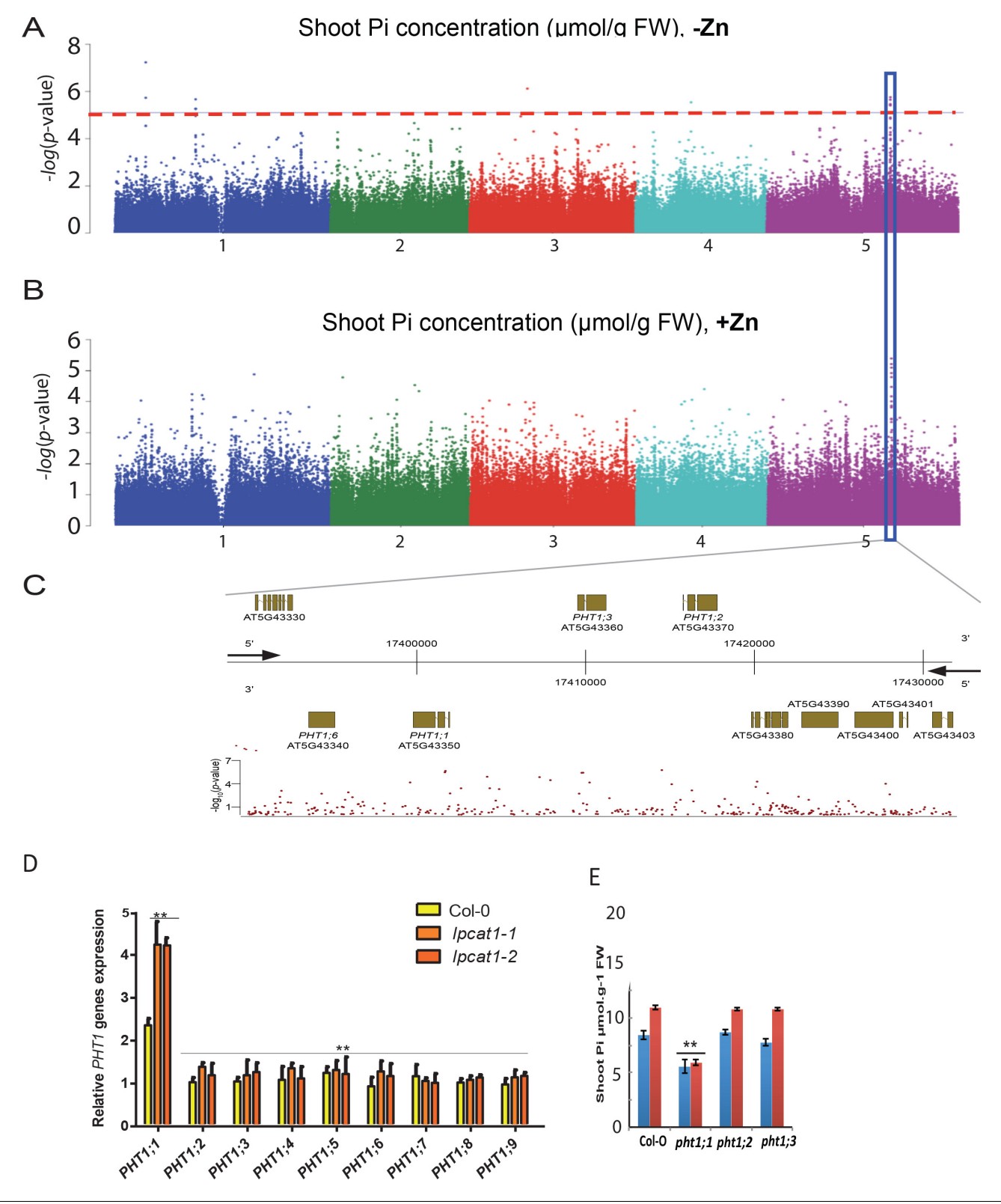

**Figure 7.** Loss of function mutations of *LPCAT1* show enhanced expression of *PHT1;1* when compared to Col-0 wild-type plants. (**A, B**) Genome-wide association (GWA) analysis of Arabidopsis shoot Pi concentration. 223 Arabidopsis thaliana accessions were grown on agar medium supplemented with zinc (+Zn) or without zinc (-Zn) for 18 days under long day conditions, upon which shoot inorganic phosphate (Pi) concentrations were determined. Manhattan plots of GWA analysis of Arabidopsis shoot Pi concentration in -Zn (**A**) and +Zn (**B**). The five Arabidopsis chromosomes are indicated in

*Figure 7 continued on next page*

*Figure 7 continued*

different colours. Each dot represents the –log10(P) association score of one single nucleotide polymorphism (SNP). The dashed red line denotes an approximate false discovery rate 10% threshold. Boxes indicate the location of the PHT1 (blue) quantitative trait loci (QTL). (C) Gene models (upper panel) and SNP –log10(P) scores (lower panel) in the genomic region surrounding the GWA QTL at the or PHT1 locus; 5' and 3' indicate the different genomic DNA strands and orientation of the respective gene models. (D) Relative expression level of all members of the Arabidopsis *PHT1* gene family in shoots of 18-days-old Col-0 wild-type plants and *lpcat1* mutants grown on -Zn agar medium compared to their expression on +Zn. mRNA accumulation was quantified by RT-qPCR, normalized to the mRNA level of the *UBIQUITIN10* reference gene (*UBQ10*: At4g05320) and expressed as relative values against its *UBQ10* normalized mRNA level of Col-0 grown in +Zn medium (control). (E) Shoot Pi concentration of 18-days-old Col-0 wild-type plants, *pht1;1*, *pht1;2*, and *pht1;3* mutants grown in +Zn or –Zn conditions. Data are mean ±SD of three biological replicates. Statistically significant differences (ANOVA and Tukey test, p<0.05 and p<0.01) are indicated by one or two asterisks.

DOI: https://doi.org/10.7554/eLife.32077.012

The following figure supplements are available for figure 7:

**Figure supplement 1.** Relative expression level of all members of the Arabidopsis *PHT1* gene family.
DOI: https://doi.org/10.7554/eLife.32077.013

**Figure supplement 2.** *High Affinity of Phosphate Transporter* (*PHT1;1*) gene expression analysis.
DOI: https://doi.org/10.7554/eLife.32077.014

observed with either *bzip19* or *bzip23* single mutants (*Assunção et al., 2010*). This redundancy may not be absolute, as recent physiological and genetic evidence indicates that bZIP19 and bZIP23 are not completely redundant and they not only regulate the same, but also separate sets of genes in Arabidopsis (*Inaba et al., 2015*). Our results support this finding by showing that only bZIP23 is involved in regulating *LPCAT1* in response to -Zn. bZIP23 is likely to do so through two *cis*-elements in the non-coding part of the *LPCAT1* gene. One being the aforementioned ZDRE, which can also be bound by the bZIP19 paralogue of bZIP23, the other a novel binding motif, with versions TGTC<u>ACA</u> and TGTC<u>GAA</u>, which are specifically bound by bZIP23. Worth noting, in the accessions panel used in our work one other allele can be found, in the Alc-0 accession (TGTC<u>AAA</u>) that displayed the lowest Pi accumulation in our panel of accession (*Supplementary file 3*, *Figure 3—figure supplement 1*). Alc-0 is the only accession among all Arabidopsis accessions for which sequence information is available in the 1001 genome database, with this version of the ZDRE motif.

The new ZDRE motif resides in the 5'-untranslated leader of *LPCAT1*. Binding of bZIP23 to this element therefore might physically block the transcription of the *LPCAT1* gene under Zn deficient conditions. This is further supported by the repressive role for bZIP23 on the expression of *LPCAT1* under Zn deficiency. Genomic sequence surveys screening for this new TF-binding site promise to further help identifying a complete list of genes potentially regulated by bZIP23 in order to fully understand the involvement of bZIP23 in the -Zn response in a genome-wide manner.

Based on our results, we propose that the sequence change in the 5'UTR sequence impacts gene (s) expression, and consequently causes variation in the associated traits. It has been proposed that evolution of species complexity from lower organisms to higher organisms is accompanied by an increase in the regulatory complexity of 5'UTRs (*Liu et al., 2012*). Nevertheless, so far little is known about the role of the 5'UTR sequence in the regulation of gene expression at the transcriptional level. The role of 5'UTR in the regulation of gene expression is perhaps best studied in human. Several studies showed that point substitutions in the 5'UTR change the expression of several genes such as *Ankirin Repeat Domain 26* (*Pippucci et al., 2011*), *Solute Carrier Family 2 Member 4* (*Malodobra-Mazur et al., 2016*), and *Cyclin D1* (*Berardi et al., 2003*). In contrast, few examples exist on the role of 5'UTR as *cis* regulators in plants. In Arabidopsis, it has been shown that the regulation of *NRP1* gene expression involves WRKY DNA binding proteins, which act on a potential *cis*-acting regulatory element located within the 5'UTR of *NRP1* (*Yu et al., 2001*). More general, a large-scale study of DNA affinity purification sequencing, which is a high-throughput TF -binding site discovery method, has revealed a global preference across TF families for enrichment at promoters and 5'UTRs (*O'Malley et al., 2016*). Taken together, our study has uncovered a new example for the role of the 5'UTR in gene expression regulation and evolution in plants.

Of a particular interest is that our study revealed that this novel bZIP23-interacting sequence motif is subject to natural variation in *A. thaliana* (*Figure 3*), and its alteration may be associated with changes in the binding capacity of bZIP23. There are several ways that genetic variants can mechanistically contribute to plant adaptation. Many reported examples with regards to nutrient

accumulation involve a change in the coding sequence of a gene that then alters the amino acid sequence of the encoded protein, thus leading to the disruption of gene function and a phenotypic change (*Baxter et al., 2010*; *Chao et al., 2012*; *Chao et al., 2014*). Reports on the role of specific regulatory element polymorphisms in the regulation of complex traits such as nutrient homeostasis crosstalk are less common, also because it is difficult to identify these relevant sequence changes. In our study, we demonstrated that allelic variation (SNPs) in the novel bZIP23 binding motif upstream of the *LPCAT1* gene is associated with variation in *LPCAT1* expression levels, which in turn results in variation in Pi accumulation in –Zn conditions. The *LPCAT1* natural variants such as found in this study offer new inspiration for agronomical and biotechnological applications to optimize Pi use efficiency in plants.

While mutation of *LPCAT1* results in altered Lyso-PC and PC concentrations; an altered Lyso-PC/PC ratio; the up-regulation of *LPCAT1* in *bzip23* mutant background has no significant effect on this ratio. The simplest explanation for this observation would be that in the *bzip23* mutant background a portion of the synthesized PC is fed into the Kennedy pathway leading to the biosynthesis of different molecules such diacylglycerol or phosphatidic acid, and therefore no changes in Lyso-PC/PC could be detected (*Wang et al., 2012*). Nevertheless, an attractive second explanation exists: LPCAT1 could be subjected to a regulation at the protein level as a strategy to optimize phosphatidylcholine specie levels. In animals, it has been proposed that the LPCAT1 primary protein sequence may contain additional motifs, structural features, or interact with second messengers or ligands that can, under certain circumstances, alter its protein half-life (*Zou et al., 2011*). The precise mechanism that regulates LPCAT1 protein stability by the ubiquitin-proteasomal pathway was already shown (*Zou et al., 2011*). Further studies will be required to verify the presence of such regulation pathway for LPCAT1 in Arabidopsis, and if confirmed, such a mechanism would add another level of our understanding of LPCAT1 activity in plants.

Mutation of *LPCAT1* results in increased *PHT1;1* expression levels (*Figure 7D*); and ultimately an over-accumulation of Pi under Zn deficiency. The induction of the expression of genes encoding P uptake transporters under Zn deficiency has been reported in crop plants such as barley (*Hordeum vulgare*) (*Huang et al., 2000*); and the model plant Arabidopsis (*Jain et al., 2013*; *Khan et al., 2014*; *Pal et al., 2017*). Our study showed an induction of *PHT1;1* in *lpcat1* plants grown under Zn deficiency. The increase in *PHT1;1* expression levels is likely to explain the increased shoot Pi concentration in *lpcat1* since it is known that CaMV 35S promoter driven overexpression of this Pi transporter significantly increases shoot Pi concentration (*Mitsukawa et al., 1997*; *Shin et al., 2004*; *Catarecha et al., 2007*). Moreover, our finding provides evidence supporting a role for a Lyso-PC/PC-derived signal in regulating Pi homeostasis under –Zn. Until recently our knowledge on PL-derived signals in plants was scarce; however, physiological and molecular studies have shown that some PL classes could serve as precursors for the generation of diverse signalling molecules (*Spector and Yorek, 1985*; *Testerink and Munnik, 2005*). For instance, Lyso-PC was shown to act as a signal for the regulation of the expression of arbuscular mycorrhiza (AM)-specific Pi transporter genes in potato, tomato and recently in *Lotus japonicus* (*Drissner et al., 2007*; *Vijayakumar et al., 2016*). In addition to the involvement of individual PLs in specific physiological processes in plants (e.g ion transport), a broader importance of changes in Lyso-PC/PC ratio for the regulation of plant development and basic cell biology is emerging. For instance, in Arabidopsis alteration of the Lyso-PC/PC ratio shortens the time to flower (*Nakamura et al., 2014*). In human cells, the Lyso-PC/PC ratio was also associated with an impairment of cell function, signalling and metabolism (*Mulder et al., 2003*; *Klavins et al., 2015*). Our data now demonstrate a fundamental link between PL metabolism, particularly Lyso-PC/PC, and Pi accumulation in –Zn condition, and lays the foundation for exploring the role of Lyso-PC/PC-derived signal in controlling ion homeostasis and response to environmental changes not only in plant cells but also in other organisms.

Overall, our study sheds light on molecular mechanism underlying an old observation made as early as 1970 s, namely P-Zn interaction in plants (*Warnock, 1970*; *Marschner and Schropp, 1977*; *Loneragan et al., 1979*). By combining GWAS and functional genomics approaches, we discovered a complete pathway involved in the regulation of shoot Pi accumulation in –Zn that can be defined as bZIP23-LPCAT1(Lyso-PC/PC)-PHT1;1. Beyond its fundamental importance, our study could have a direct impact on plant growth in field by improving plant growth while reducing P supply, and will help meeting one of challenges facing agriculture in the 21st century.

## Materials and methods

### Plant materials and growth conditions

A subset of 223 *Arabidopsis thaliana* accessions of the RegMap panel (*Horton et al., 2012*) was used for genome-wide association studies (Arabidopsis Biological Resource Center accession number CS77400). The names of accessions are provided in *Supplementary file 1*. All lines were used side by side in the same growth chambers under the same conditions, 22°C under long days (16 hr light and 8 hr dark). Arabidopsis mutants used in this study are in the Columbia-0 genetic background. The *bzip19bzip23* mutant previously described by *Assunção et al. (2010)* was used in this work. T-DNA insertion mutant lines for the At5g43350 (N666665, *pht1;1*), At5g43360 (N661080, *pht1;2*), At5g43370 (N448417, *pht1;3*), At1g12640 (N686743 (*lpcat1-1*, [*Wang et al., 2012*]), N442842) and At1g12650 (N526222) genes were obtained from the European Arabidopsis Stock Centre (arabidopsis.info; University of Nottingham, UK). Plants were germinated and grown on vertically positioned agar-solidified media (A1296, Sigma). The complete nutrient medium contained: 9.5 mM $KNO_3$, 10.3 mM $NH_4NO_3$, 1.5 mM $MgSO_4$, 1 mM $KH_2PO_4$, 2 mM $CaCl_2$, 100 $\mu$M FeNaEDTA, 100 $\mu$M MnSO4, 30 $\mu$M $ZnSO_4$, 100 $\mu$M $H_3BO_3$, 5 $\mu$M KI, 1 $\mu$M $Na_2MoO_4$, 0.1 $\mu$M $CuSO_4$ and 0.1 $\mu$M $CoCl_2$ (adapted from *Murashige and Skoog, 1962*). Zn-deficient medium was made by omitting $ZnSO_4$. Seeds sown on plates were stratified at 4°C for 3 days. Plates were then transferred to a growth chamber for 18 days set at the following conditions: 16/8 hr light/dark cycle, 250 µmol m$^{-2}$ s$^{-1}$ light, and 24/20°C (light/dark).

### Plasmid construction and plant transformation

The *LPCAT1* coding region driven by its native promoter (1.5 kbp fragment immediately upstream of the start codon including the 5' untranslated region (5'-UTR)) from Col-0 and Sap-0 accessions were amplified using PCR and the following primers p*LPCAT1* $^{Col-0}$-forward 5'-cgctgcagggtgtcgaaaacccgtttt-3'; p*LPCAT1*$^{Col-0}$-*reverse* 5'-cgggatcctgatcagagagttacaac aggagag-3'; p*LPCAT1*$^{Sap-0}$-forward 5'-cgctgcagggtgtcacaaacccgggt-3' and p*LPCAT1*$^{Sap-0}$-*reverse* 5'-cgggatccatgatcagatagttacaacaggagagg-3', and then cloned into the binary vector pCAMBIA1301 by restriction enzymes *Bam*HI and *Pst*I (site underlined). The *LPCAT1* coding regions were amplified using PCR and the following primers p*LPCAT1*$^{Col-0}$-forward 5'-cgctgcagttattcttctttacgcggttttg-3'; p*LPCAT1*$^{Sap-0}$-forward 5'-cgctgcagttattcttctttacgtggttttggt-3' and p*LPCAT1*$^{Col-0/ Sap-0}$-*reverse* 5'-cgctgcagatggatatgagttcaatggctg-3'. *Pst*I was used for the fusion of p*LPCAT1*$^{Col-0}$ or p*LPCAT1*$^{Sap-0}$ promoters to either *LPCAT1*$^{Col-0}$ or *LPCAT1*$^{Sap-0}$. The constructs were transformed into *Agrobacterium tumefaciens* strain GV3101 and then used for Arabidopsis transformation by the floral dip method (*Clough and Bent, 1998*). Transgenic plants were selected by antibiotic resistance, and only homozygote descendants of hemizygote T2 plants segregating 1:3 for antibiotic resistance: sensitivity were used for analysis.

### Inorganic phosphate concentration measurements and GWA mapping

All accessions were grown in the presence or absence of zinc for 18 days. Shoots were collected, weighed and ground into powder in liquid nitrogen. An aliquot (30 mg) was incubated at 70°C in NanoPure water, for 1 hr. Inorganic phosphate (Pi) concentrations were determined using the molybdate assay as previously described by *Ames (1966)*. The shoot Pi concentrations across the analysed accessions was used as phenotype for GWA analysis. The GWA analysis was performed in the GWAPP web interface using the mixed model algorithm (AMM) that accounts for population structure (*Seren et al., 2012*) and using the SNP data from the RegMap panel (*Atwell et al., 2010*; *Brachi et al., 2010*; *Horton et al., 2012*). Only SNPs with minor allele counts greater or equal to 10 (at least 10 out of 223 accessions contained the minor allele) were taken into account. To correct for multiple testing, the false discover rate was calculated using the Benjamini-Hochberg correction (*Benjamini and Hochberg, 1995*). An FDR threshold of 0.1 was used to detect significant associations.

### Haplotype analysis

Haplotype analysis was performed as follows. SNPs from a 50 kb window around the significant marker SNP (chromosome 1, position 4306845) were extracted for the 221 natural accessions. SNP

data were taken from the Regional Mapping Project SNP panel described in *Horton et al. (2012)*. These SNPs were used as the input for fastPHASE version 1.4.0 (*Scheet and Stephens, 2006*). Results were then visualized using R.

## Gene expression analysis by quantitative RT-PCR

For expression analysis, the Plant RNeasy extraction kit (Qiagen) was used to extract total RNA free of residual genomic DNA from 100 mg frozen shoot material. Total RNA was quantified with a Nano-Drop spectrophotometer (Thermo Scientific).Two $\mu$g of total RNA was used to synthesize cDNA. Reverse transcriptase PCR (RT-qPCR) was performed with a Light Cycler 480 Real-Time PCR System (Roche) using SYBR green dye technology (Roche) as described by *Khan et al. (2014)*. The primers used in this study are *LPCAT1*-forward 5'-ggtgttaagcttgcacgaaac-3'; *LPCAT1*-reverse 5'-agagaaacaa-gaaccgga-3' and *UBQ10*-forward 5'-aggatggcagaactcttgct-3'; *UBQ10*-reverse. 5'-tcccagtcaacgtct-taacg-3'. The primers used to quantify *ZIP4* are *ZIP4*-forward 5'-cggttaaacataagaaatcaggagc-3'; *ZIP4*-reverse 5'-taaatctcgagcgttgtgatg-3'; and for *ZIP12* are *ZIP12*-forward 5'-aacagatctcgcttggcg-3'; *ZIP12*-reverse 5'-aatgtgatcatcatcttggg-3'. Primers used to quantify the *PHT1* gene family member are designed according to *Khan et al. (2014)*. Quantification of mRNA abundance was performed in a final volume of 20 µL containing 10 µL of the SYBR Green I master mix, 0,3 µmol primers, and 5 µL of a 1:25 cDNA dilution. PCR conditions were as 95℃ for 5 min, and followed by 40 cycles of 95℃ for 10 s, 60℃ for 10 s, 72℃ for 25 s. One final cycle was added in this program: 72℃ for 5 min. For every reaction, the cycle threshold (Ct) value was calculated from the amplification curves. For each gene, the relative amount of calculated mRNA was normalized to the calculated mRNA level of the *Ubiquitin10* control gene (*UBQ10*: At4g05320) and expressed as relative values against wild-type plants grown in the presence or absence of Zn in the medium. Quantification of the relative transcript levels was as described in *Rouached et al. (2008)*. The mRNA abundance of each gene was calculated following normalization against the CT values of *Ubiquitin10*mRNA, for instance $\Delta$Ct, *LPCAT1* = Ct,*LPCAT1* − (Ct,*UBQ10*). Quantification of the relative transcript levels was performed as following, low Zn (-Zn) treatment was compared to control (Ct,+Zn), the relative mRNA accumulation of each gene was expressed as a $\Delta\Delta$Ct value calculated as follows: $\Delta\Delta$Ct = $\Delta$Ct, *LPCAT1*(-Zn) − $\Delta$Ct,*LPCAT1*(Ct). The fold change in relative gene expression was determined as $2^{-\Delta\Delta Ct}$.

## Expression and purification of bZIP19 and bZIP23 proteins

*bZIP19 and bZIP23* coding sequences CDS were first cloned in the pENTR/D-TOPO vector, and then transferred to pDEST15 vector (Invitrogen) by LR reaction following the manufacturer's instructions. The GST-bZIP19 and GST-bZIP23 fusion proteins were expressed in *Escherichia coli* Rosetta 2(DE3) pLysS (Novagen, Darmstadt, Germany). Transformed cells were grown in a phosphate-buffered rich medium (Terrific broth) at 37℃ containing appropriate antibiotics until the OD$_{660}$ reached 0.7–0.8. After induction with 1 mM IPTG (isopropyl-b-D-thiogalactoside) for 16 hr at 22° C, bacteria were harvested by centrifugation (6000 $\times$ *g*, 10 min, 4℃) and suspended in 1X PBS buffer containing lysozyme from chicken egg white (Sigma) and complete protease inhibitor cocktail (Roche). The resulting cell suspension was sonicated and centrifuged at 15,000 $\times$ g, for 15 min at 4℃ to remove intact cells and debris. The protein extract was mixed with buffered glutathione sepharose beads (GE Healthcare, Freiburg, Germany), and incubated at 4℃ for 3 hr. The resin was centrifuged (500 $\times$ *g*, 10 min, 4℃) and washed five times with 1X PBS buffer.

bZIP19 and bZIP23 were then cleaved from GST using 25 unit/ml of thrombin at room temperature for 16 hr. All fractions were subjected to SDS-PAGE, and protein concentrations were determined. For protein quantification, absorbance measurements were recorded on a nanodrop spectrophotometer (Model No.1000, Thermo Scientific Inc., Wilmington, Delaware, USA) at 280 nm, and in parallel on a VICTOR2 microplate reader (MULTILABEL COUNTER, life sciences) at 660 nm using the Pierce 660 nm Protein Assay (Pierce/Thermo Scientific, Rockford; [*Antharavally et al., 2009*]).

## Electophoretic mobility shift assay (EMSA)

EMSA was performed using purified proteins and DNA probes labeled with Biotin-TEG at the 3' end. Biotin-TEG 3' end-labeled single-stranded DNA oligonucleotides were incubated at 95℃ for 10 min and then annealed to generate double-stranded DNA probes by slow cooling. The sequences

of the oligonucleotide probes were synthesized by Eurofins Genomics and are as following: 5′-ttaggttcac**gtgtcgacat**gaaaggagct-3′, 5′-catatccatg**gtgtcgaa**aacccgattttt-3′ and 5′-catatccatg**gtgtca-ca**aacccgggtttt-3. .The binding of the purified proteins (≈ 150 ng) to the Biotin-TEG labelled probes (20 fmol) was carried out using the LightShift Chemiluminescent EMSA Kit (Thermo Scientific, Waltham, USA) in 20 µL reaction mixture containing 1X binding buffer (10 mM Tris, 50 mM KCl, 1 mM DTT, pH 7.5), 2.5% glycerol, 5 mM MgCl$_2$, 2 µg of poly (dI-dC) and 0.05% NP-40. After incubation at 24° C for 30 min, the protein–probe mixture was separated in a 4% polyacrylamide native gel at 100 V for 50 min then transferred to a Biodyne B Nylon membrane (Thermo Scientific) by capillary action in 20X SSC buffer overnight. After ultraviolet crosslinking (254 nm) for 90 s at 120 mJ.cm$^{-2}$. The migration of Biotin-TEG labelled probes was detected using horseradish peroxidase-conjugated streptavidin in the LightShift Chemiluminescent EMSA Kit (Thermo Scientific) according to the manufacturer's protocol, and then exposed to X-ray film.

## Phospholipid extraction

Lipids were extracted from 18-days-old *Arabidopsis thaliana* shoots (Col-0) grown in the presence or absence of Zn, following the Folch's method (*Folch et al., 1957*). The total phosphorus (P) contained in lipids was measured using a spectrophotometer with an absorbance at 830 nm. Lipid separation and quantification was performed using Thin Layer Chromatography (TLC). The lipid composition was detected and quantified using a GAMAG TLC SCANNER 3 (Muttenz, Switzerland), operating in the reflectance mode. The plates were scanned at 715 nm after dipping in a solution of Blue Spray (Sigma, France) and heating for 3 min at 55°C. The WinCat software program was used to scan bands, the different classes of phospholipids (*Fouret et al., 2015*) were identified by comparing their retention factor (Rf) to authentic standards and the quantities of each phospholipid were evaluated against the corresponding calibration curve (*Fouret et al., 2015*).

## *In planta* transactivation assay

The *in planta* transactivation assay was performed in *N. benthamiana*. LPCAT Col-0 promoter and LPCAT Col-0 mutated promoter were cloned pLPCAT1Col-0-For 5′-ggggacaagtttgtacaaaaaag-caggcttc**ggtgtcgaaaacccgttt**-3′; pLPCAT1Col-0-Rev 5′- ggggaccactttgtacaagaaagctgggtctgatca-gagttacaacaggagag −3′; pLPCAT1Col-0-mutated-For 5′-ggggacaagtttgtacaaaaaagcaggcttcggtgtcacaaacccgtttt-3′. The LPCAT1 promoters were then fused to the β-GUS-encoding reporter gene using the Gateway system. These following primers were used to clone the promoter of the zinc transporter ZIP4-For 5′- ggggacaagtttgtacaaaaaagcaggcttc**ttg-gaaagtgaagtggattg**-3′; ZIP4-REV 5′- ggggaccactttgtacaagaaagctgggtctgatc**atcgacgaagaccatgg-gaacaagagt** −3′ (*Lin et al., 2016*). Each of these primer was then fused to β-GUS-encoding reporter gene. bZIP23 coding sequences CDS was placed under the CaMV. 35S promoter. The 35S::C-YFP construct was provided by Dr. Seung Y. Rhee (*Bossi et al., 2017*). Each construct was transformed into Agrobacterium. Positive clones per construct were grown overnight at 28°C, and then washed four times in infiltration buffer (10 mM MgCl$_2$, 10 mM MES (pH 5.6) and 100 uM acetosyringone). The effector construct and reporter construct (OD600 of 0.8), were co-infiltrated at a ratio of 9 to 1 in fully expanded leaves (third or fourth leaves) of five to six week-old tobacco plants. Each plasmid combination was infiltrated in one leaf from different plants. We used C-YFP as negative controls. Three independent infiltrations per combination resulting in six samples per construct. Three days after infiltration, leaves were collected, and the infiltrated areas in each leaf were excised and pooled into one sample. The GUS extraction and the GUS enzymatic activity measurements was performed as described by *Bossi et al., 2017*. The relative GUS enzymatic activity was determined by comparing the effect of bZIP23 TF and C-YFP protein on each promoter.

## Statistical analysis

Statistical analysis of quantitative data was performed using the GraphPad prism 5.01 software program for Windows (GraphPad 156 Software, CA, USA, http://www.graphpad.com). For all the t-test analyses the difference was considered statistically significant when the test yielded a p-value <0.05.

## Acknowledgement

The authors are grateful to Dr. Santosh B Satbhai and Bonnie Wohlrab for initial seed preparation of accessions and to Christian Goeschl for help with the Manhattan plots, to Drs Jérôme Lecomte and Christine Feillet-Coudray for their help with the lipid quantification. Thanks to Prof Pierre Berthomieu, Drs Patrick Doumas, Saber Kouas and Zaigham Shahzad for helpful discussions. Thanks to Dr Seung Y Rhee LAB member (Carnegie Institution for Science, Stanford, USA) for help with the *in planta* transactivation assay. This work was funded by the Institut National de la Recherche Agronomique (INRA) and by the Région Languedoc-Roussillon: Chercheur d'Avenir 2015, Projet cofinancé par le Fonds Européen de Développement Régional to HR, the Austrian Academy of Sciences through the Gregor Mendel Institute and the Salk Institute for Biological Studies to WB, the Netherlands Genome Initiative ZonMW Horizon program Zenith project no. 40-41009-98-11084 supporting MA and RA, and by an Iraq government doctoral fellowship for MK.

## Additional information

### Funding

| Funder | Grant reference number | Author |
|---|---|---|
| Institut National de la Recherche Agronomique | | Hatem Rouached |
| The Austrian Academy of Sciences through Gregor Mendel Institure | | Wolfgang Busch |
| ZonMw | no. 40-41009-98-11084 | Mark GM Aarts |
| Iraq Government Doctoral Fellowship | | Mushtak Kisko |
| Région Languedoc-Roussillon: Chercheur d'Avenir/Projet Cofinancé par le Fonds Européen de Développement Regional | | Hatem Rouached |

The funders had no role in study design, data collection and interpretation, or the decision to submit the work for publication.

### Author contributions

Mushtak Kisko, Data curation, Formal analysis, Investigation, Methodology; Nadia Bouain, David Secco, Formal analysis, Investigation, Methodology; Alaeddine Safi, Anna Medici, Gilles Fouret, Gabriel Krouk, Formal analysis, Methodology; Robert C Akkers, Resources, Formal analysis, Methodology, Writing—original draft; Mark GM Aarts, Resources, Formal analysis, Funding acquisition, Writing—original draft; Wolfgang Busch, Resources, Formal analysis, Validation, Methodology, Writing—original draft; Hatem Rouached, Conceptualization, Resources, Formal analysis, Supervision, Funding acquisition, Validation, Writing—original draft, Project administration

### Author ORCIDs

Alaeddine Safi  http://orcid.org/0000-0003-1532-5708
Hatem Rouached  http://orcid.org/0000-0002-7751-0477

### Decision letter and Author response

Decision letter https://doi.org/10.7554/eLife.32077.020
Author response https://doi.org/10.7554/eLife.32077.021

## Additional files

**Supplementary files**

• Supplementary file 1. Shoots Pi concentration in the 223 *Arabidopsis thaliana* accessions grown in presence or absence of zinc.

DOI: https://doi.org/10.7554/eLife.32077.015

• Supplementary file 2. Coordinates for the significant SNPs associated with Pi concentration in shoots in Zn conditions.

DOI: https://doi.org/10.7554/eLife.32077.016

• Supplementary file 3. List of new ZDRE motif in the *Arabidopsis thaliana* accessions and the shoot Pi content in presence (+Zn) or absence of Zn (-Zn).

DOI: https://doi.org/10.7554/eLife.32077.017

• Transparent reporting form

DOI: https://doi.org/10.7554/eLife.32077.018

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
