## [Decision Letter]

Thank you for submitting your article "*LPCAT1* controls phosphate homeostasis in a zinc-dependent manner" for consideration by *eLife*. Your article has been favorably evaluated by Detlef Weigel (Senior Editor) and three reviewers, one of whom is a member of our Board of Reviewing Editors. The following individual involved in review of your submission has agreed to reveal his identity: Daniel J Kliebenstein (Reviewer #2).

The reviewers have discussed the reviews with one another and the Reviewing Editor has drafted this decision to frame the key concerns we feel must be addressed before a binding decision may be made on your submission.

Summary:

In plants, regulation of phosphate accumulation in shoots is disrupted under conditions of Zn deficiency with the result that plant over-accumulate Pi. Here, the authors provide the first insights the molecular basis of cross-talk between Zn and Pi signaling pathways. Through a GWA study and subsequent analyses of mutants, the authors identify variation in the *LPCAT1* gene expression as the cause of variation in Zn deficiency-induced Pi accumulation. A new zinc deficiency cis-element upstream of the *LPCAT1* coding sequence and the bZIP23 transcription factor are proposed cause the variation in *LPCAT1* expression which in turn alters LPC/PC ratios and expression of phosphate transporter genes. The reviewers felt that these data, if fully substantiated, provide evidence for a causal relationship between cis-regulatory variation and Pi accumulation. The reviewers propose that the PHR data be omitted as currently, there is no evidence for a link between PHR and *LPCAT1*. The reviewers consider the following revisions necessary to fully substantiate the main claims of the manuscript and to improve the overall presentation.

Essential revisions:

1) Promoter swap experiments (Figure 4). The statistical analysis in the Figure 4 does not match the claims in the text. Figure 4 shows comparisons only to Col. It is necessary to demonstrate whether there is a difference between the variants and the empty vector complemented *LPCAT1*, not only Col. The best analysis would be to do ANOVA with just the promoter swap lines and to directly test how the promoter or the ORF effects the phenotype. Col-0 is not the proper control given that the lines are in the *lpcat* background. This can be accomplished by making an ANOVA that goes Pi = promoter + ORF + promoter x ORF and then the authors could get direct p-values for the key terms that they wish to report.

2) Do the complemented lines shown in Figure 4 (*lcpat1* mutant expressing the different *LPCAT1* variants) show changes in LPC and PC content. The claim that the regulatory element is responsible for the alterations in P content and that this occurs by alterations in LPC/PC ratio should be supported with LPC/PC ratio data for these lines.

3) The promoter swap experiments (Figure 4) do not demonstrate that the effect is due to the new ZDRE motif. It could be due to additional allelic variation in other motifs. In the first paragraph of the subsection “Allelic variation of *LPCAT1* determines natural variation of Pi content under zinc deficiency.”, the authors indicate that there is no variation in the common ZDRE but they do not mention whether there is additional variation in other potential motifs within the promoter that might correlate with shoot Pi content in -Zn (it is possible as Col-0 and Sap-0 promoters displayed 97,9% sequence identity. Therefore, there are additional SNPs). To demonstrate that the new ZDRE is responsible for Zn dependent control of *LPCAT1* expression, mutant versions of the ZDRE motif should be included in the promoter swap experiments.

4) The authors find allelic variation in the sequence of the newly identified Zn-deficiency responsive element (ZDRE) that correlates with variation in the shoot Pi content of contrasting groups of accessions. However, whether other polymorphisms exist in the same motif for other accessions is not mentioned. This information should be provided and if indeed polymorphisms do exist, this must be discussed and the effects on *LPCAT1* gene expression and shoot Pi data should be added.

5) Figure 2 (element binding analysis). This analysis is missing a negative control. Also, the important question of differential binding to the ZDRE variants is not addressed. Additional EMSA analyses that evaluate the binding affinity of bZIP23 to the two sequence variants of the new ZDRE in the *LPCAT1* promoter must be included. Given that this new element is considered to be responsible for the differential expression of the *LPCAT1* gene, an assessment of differential binding (through competition assays) is essential.

It is also possible that bZIP19 binds the ZDRE variant found in accessions accumulating more Pi in the shoot under -Zn (e.g. GTGTCACA). Such a mechanism may help to further repress *LPCAT1* expression under -Zn. Indeed, combinatorial effects between the two ZDRE found in the *LPCAT1* promoter affecting *LPCAT1* gene expression should not be discounted. If necessary, further assessment of bZIP19 binding should be considered.

6) Figure 5. *lpcat1* mutants display increased lyso-PC/PC ratio. However, *bzip23* mutation, which results in increased *LPCAT1* expression, has no significant effect on this ratio. Discussion should be provided to explain this difference that affects the proposed model. Related to this, is there any significant difference in lyso-PC/PC ratio between accessions showing contrasting shoot Pi content in -Zn?

7) With regard to *LPCAT1*, a SNP explaining 11% of shoot Pi variation in -Zn was found. However, details on the contribution of the *PHT1* loci to this trait are not provided. A more detailed description of the GWAS results must be provided for the latter (significant SNPs specific of -Zn in all *PHT1*s?).

8) How sure are the authors that there are only two causal alleles of *LPCAT1* (Col-0/SAP). This seems to be an implicit claim but rarely are causal genes dimorphic in Arabidopsis and more often they have multiple alleles. This does not take away from the causation they have found but they should definitely be clear that there may be other alleles in the text or they could conduct a phylogenetic analysis of the promoter and ORF to classify their alleles within a broader context.

9) In several cases the text requires modification to improve accuracy. Please modify the text to address the following points.

a) The authors should be a bit careful in their claims. For example, in the subsection “GWAS identify two candidate genes involved in the accumulation of Pi in the shoot under Zn deficiency”, this RT-PCR shows that Col-0 is responding to the Zn limitation but doesn't inherently mean that all accessions are responding in the same way or even at all to the Zn limitation. It is a nice control but is only a control for Col-0.

b) Similarly, in the subsection “GWAS identify two candidate genes involved in the accumulation of Pi in the shoot under Zn deficiency”, yes the heritability is high but the fact that there is a difference in the presence or absence of Zn means that this is really a G X E trait and not largely genetic. To make the largely genetic claim would require a G + E + G x E analysis to partition the variance.

c) In the subsection “GWAS identify two candidate genes involved in the accumulation of Pi in the shoot under Zn deficiency”, SNP effect estimates are notoriously difficult to make accurate so I would suggest caution in using 11% vs. 49% to say this gene is the main effect. The Manhattan plots suggest that there are likely a number of other loci and if there is trans-LD between these loci as found by Brachi and Bergelson for defense metabolism, then individual SNP estimates are inflated as they capture the causation at other SNPs. This in no way diminishes the fact that they have found a gene involved. It is just better to be accurate in claims than to inflate claims.

d) Sometimes, the logic of the experimental decisions was hidden in the text. For example, in the first paragraph of the subsection “*LPCAT1* acts downstream of bZIP23 transcription factor”, the authors find a ZDRE in *LPCAT1* which helps to support the GWA association. Yet this section is introduced by saying that the goal was to understand the regulatory context which is a very general goal that implies the section will discuss all the promoter elements. It seems that if the focus is solely on ZDRE, that it would make sense to state that the goal was to provide more molecular causational links between Zn/LPCAT/Pi which hasn't really been well understood.

10) Some important details are missing from the data and some figures are incorrect and/or the presentation is poor. Please address these points:

a) The coordinates for the significant SNP associated with Pi concentration in shoots in Zn- conditions must be provided.

b) The two motifs (shown in Figure 2) are on the opposite strand from the ATG, so that figure is incorrect. The figure should be corrected.

c) Figure 1. Please plot the data rather than showing histograms.

d) Figure 6. The legend indicates that expression in both high and low Zn is shown but this is not the case. Please show the relative expression in both conditions.

e) An additional supplementary table listing the sequence of the newly identified ZDRE in all accessions analyzed should be included to make correlations with their shoot Pi content in +Zn/-Zn.

11) Please consider the following comment which may help strengthen the PHT analyses:

For the PHT analyses, it seems odd to rely on a Bonferroni when the authors have direct evidence that this is likely too conservative and causing false negative results as shown from the PHT1 data. There are plenty of papers showing that Bonferroni is overly conservative. I would suggest using a more realistic p-value and then the authors could bring in the PHT transporters as a part of the GWA causal network. Right now it has an artificial spurious argument of saying "they weren't significant but almost were". An option would be to use the modified Bonferroni approach where you estimate the true number of independent SNPs within a dataset and use that as the denominator rather than the total number of SNPs which is really inflates the correction.

12) It is surprising that you have not used 1001 Genomes imputed SNPs for your GWAS. Please rerun the analysis and report whether the apparently causal SNP is among the best hits. Consultation of the 1001 Genomes website indicates that this SNP is well represented in the 1001 Genomes data, with ~80% accessions having the standard allele, ~10% the alternative allele, and ~10% missing data. You should also show directly the Pi data for the accessions you analyzed, comparing distribution of Pi levels in all of your accessions with the standard versus all accessions with the alternative allele.

13) Please give the genome coordinates for all sequences discussed in the paper, especially the putatively causal SNP. Note also that Figure 2 has a significant error, as the ATG and the two cis motifs are on opposite strands. Please correct. In addition, please discuss that the apparently causal motif is immediately upstream of the ATG and in the 5' UTR. Briefly discuss what is known about transcription factors binding in the 5' UTR and acting as cis-regulators.

---

## [Author Response]

Summary:In plants, regulation of phosphate accumulation in shoots is disrupted under conditions of Zn deficiency with the result that plant over-accumulate Pi. Here, the authors provide the first insights the molecular basis of cross-talk between Zn and Pi signaling pathways. Through a GWA study and subsequent analyses of mutants, the authors identify variation in the *LPCAT1* gene expression as the cause of variation in Zn deficiency-induced Pi accumulation. A new zinc deficiency cis-element upstream of the *LPCAT1* coding sequence and the bZIP23 transcription factor are proposed cause the variation in *LPCAT1* expression which in turn alters LPC/PC ratios and expression of phosphate transporter genes. The reviewers felt that these data, if fully substantiated, provide evidence for a causal relationship between cis-regulatory variation and Pi accumulation. The reviewers propose that the PHR data be omitted as currently, there is no evidence for a link between PHR and *LPCAT1*. The reviewers consider the following revisions necessary to fully substantiate the main claims of the manuscript and to improve the overall presentation.

PHR1 was omitted as suggested by the reviewer.

Essential revisions:1) Promoter swap experiments (Figure 4). The statistical analysis in the Figure 4 does not match the claims in the text. Figure 4 shows comparisons only to Col. It is necessary to demonstrate whether there is a difference between the variants and the empty vector complemented LPCAT1, not only Col. The best analysis would be to do ANOVA with just the promoter swap lines and to directly test how the promoter or the ORF effects the phenotype. Col-0 is not the proper control given that the lines are in the lpcat background. This can be accomplished by making an ANOVA that goes Pi = promoter + ORF + promoter x ORF and then the authors could get direct p-values for the key terms that they wish to report.

As suggested by the reviewer, we performed two-way ANOVA analysis and results reported as table in Figure 4. The analysis confirmed our claim that Pi accumulation is associated with promoter changes.

2) Do the complemented lines shown in Figure 4 (lcpat1 mutant expressing the different LPCAT1 variants) show changes in LPC and PC content. The claim that the regulatory element is responsible for the alterations in P content and that this occurs by alterations in LPC/PC ratio should be supported with LPC/PC ratio data for these lines.

As recommended by the reviewer we tested whether the polymorphisms in the regulatory region of *LPCAT1* are responsible for the change in LPC/PC ratio that ultimately affect the Pi content in –Zn conditions. We determined the LPC, PC concentrations in the shoots of the plants expressing *LPCAT1* driven by LPCAT1^Col-0^ promoter (pLPCAT1^Col-0^::*LPCAT1*^Col-0^, pLPCAT1^Col-0^::*LPCAT1*^Sap-0^) or LPCAT1^Sap-0^ promoter (pLPCAT1^Sap-0^ ::*LPCAT1*^Col-0^, pLPCAT1^Sap-0^::*LPCAT1*^Sap-0^) in the *lpcat1* mutant background, grown in the presence or absence of Zn for 18 days. WT plants of the Col-0, Sap-0 accessions, and *lpcat1* transformed with the empty vector were included in this analysis. In the presence of Zn no difference in PC or LPCA concentrations was observed between all plant lines. However, under –Zn conditions, Sap-0, pLPCAT1^Sap-0^::*LPCAT1*^Col-0^, pLPCAT1^Sap-0^::*LPCAT1*^Sap-0^ or *lpcat1* lines showed a significant decrease of PC and increase of LPC concentrations respectively, leading to an increase of Lyso-PC/PC ratios (Figure 6). These results further support the association between the increase of LPC/PC ratios and the alterations in P content in the plant shoots under Zn deficiency (Figure 4). These new results are presented in Figure 6.

3) The promoter swap experiments (Figure 4) do not demonstrate that the effect is due to the new ZDRE motif. It could be due to additional allelic variation in other motifs. In the first paragraph of the subsection “Allelic variation of LPCAT1 determines natural variation of Pi content under zinc deficiency.”, the authors indicate that there is no variation in the common ZDRE but they do not mention whether there is additional variation in other potential motifs within the promoter that might correlate with shoot Pi content in -Zn (it is possible as Col-0 and Sap-0 promoters displayed 97,9% sequence identity. Therefore, there are additional SNPs). To demonstrate that the new ZDRE is responsible for Zn dependent control of LPCAT1 expression, mutant versions of the ZDRE motif should be included in the promoter swap experiments.

To assess the effect of the bZIP23 TF on the activity of the native Col-0 *LPACT1* promoter (with « GTGTCGAA » as new ZDRE) and a modified (point mutation) version of the Col-0 *LPCAT1* promoter to only contain the new ZDRE of Sap-0 (« GTGTCACA ») we used a quantitative in planta transactivation assay as described by Bossi et al., 2017. In this assay, each *LPCAT1* promoter version (native or mutated) was fused to a β-glucuronidase (GUS)-encoding reporter gene. 35S:bZIP23 and 35S::C-YFP were used as effectors (Figure 3). The comparison of the ability of 35S-bZIP23 or 35S-YFP to activate either *LPCAT1* promoter versions was performed by quantifying the GUS activity. Each reporter construct was co-transfected with either the 35S:bZIP23 or 35S::C-YFP construct into tobacco leaves. As a positive control, we tested the effect of 35S:bZIP23 on its known target promoter of At*ZIP4* gene fused to β-GUS-encoding reporter gene (Assunção et al., 2010). As expected, our results showed an induction of the activity of At*ZIP4* promoter by 35S:bZIP23 (positive control). More importantly, our results showed that bZIP23 represses the activity of the *LPCAT1* promoter that contains the new ZDRE of Sap-0 compared to the Col-0 *LPACT1* native promoter. These results are consistent with our qPCR data showing that *LPCAT1* is less expressed in Sap-0 compared to Col-0 background under Zn deficiency (Figure 2). These results are now presented in Figure 3.

4) The authors find allelic variation in the sequence of the newly identified Zn-deficiency responsive element (ZDRE) that correlates with variation in the shoot Pi content of contrasting groups of accessions. However, whether other polymorphisms exist in the same motif for other accessions is not mentioned. This information should be provided and if indeed polymorphisms do exist, this must be discussed and the effects on LPCAT1 gene expression and shoot Pi data should be added.

We have used 223 accessions for our GWA studies, and focused on alleles with a minimum allele frequency (MAF) of 0.05 to test their role in shoot Pi accumulation. We used the 1001 genome database to retrieve the *LPCAT1* sequences of 163 accessions of our panel. Only 10 of these accessions have the new ZDRE motif « GTGTCACA » and 152 accessions contain the standard version of this motif « GTGTCGAA » in their promoter. There is only one other allele, found in the Alc-0 accession, that harbours a « GTGTCAAA » motif sequence. In fact, Alc-0 is the only accession among all Arabidopsis accessions for which sequence information is available in the 1001 genome database, with this version of the ZDRE motif. While the mutation in the new ZDRE motif of the Alc-0 *LPCAT1* promoter is potentially interesting, based on the current information it is not possible to determine with any statistical significance if this polymorphism affects gene expression or shoot Pi content. This finding does not at all change the message of the paper.

This information is now included in the Discussion section. We included Supplementary file 3 listing the new ZDRE motif in the accessions analyzed, with the shoot Pi content in both conditions (+Zn and –Zn).

5) Figure 2 (element binding analysis). This analysis is missing a negative control. Also, the important question of differential binding to the ZDRE variants is not addressed. Additional EMSA analyses that evaluate the binding affinity of bZIP23 to the two sequence variants of the new ZDRE in the LPCAT1 promoter must be included. Given that this new element is considered to be responsible for the differential expression of the LPCAT1 gene, an assessment of differential binding (through competition assays) is essential.It is also possible that bZIP19 binds the ZDRE variant found in accessions accumulating more Pi in the shoot under -Zn (e.g. GTGTCACA). Such a mechanism may help to further repress LPCAT1 expression under -Zn. Indeed, combinatorial effects between the two ZDRE found in the LPCAT1 promoter affecting LPCAT1 gene expression should not be discounted. If necessary, further assessment of bZIP19 binding should be considered.

We performed a new EMSA analysis to test the capacity of bZIP23 and bZIP19 to bind to the other version of the new ZDRE motif, GTGTCACA. Our results showed that only bZIP23 can bind to this version of the new ZDRE motif. The new results are presented in Figure 3. These new results further reinforce our previous EMSA data presented in Figure 2 where two positive where bZIP23 and bZIP19 bind to the already known ZDRE (RTGTCGACAY) and only bZIP23 binds to one version of the new ZDRE motif (GTGTCGAA). Taken together, our EMSA results (Figure 2, Figure 3) support the specificity of new ZDRE motif for bZIP23.

We are not aware that EMSA can provide a reliable quantitative result. As far as we know, differential EMSA is not trivial and using this in a competition mode may not work. Instead we performed the experiments as detailed in point 3 (a quantitative in planta transactivation assay) to address the reviewer’s comment.

6) Figure 5. lpcat1 mutants display increased lyso-PC/PC ratio. However, bzip23 mutation, which results in increased LPCAT1 expression, has no significant effect on this ratio. Discussion should be provided to explain this difference that affects the proposed model. Related to this, is there any significant difference in lyso-PC/PC ratio between accessions showing contrasting shoot Pi content in -Zn?

We have now added a short paragraph in the Discussion section: “While mutation of *LPCAT1* results in altered Lyso-PC and PC concentrations; an altered Lyso-PC/PC ratio; the up-regulation of *LPCAT1* in *bzip23* mutant background has no significant effect on this ratio. […] Further studies will be required to verify the presence of such regulation pathway for *LPCAT1* in Arabidopsis, and if confirmed, such a mechanism would add another level of our understanding of *LPCAT1* activity and phosphatidylcholine accumulation in plants.”

LPC, PC concentrations were analyzed and the LPC/PC ratios in Col-0 and Sap-0 (genotypes that showed contrasting shoot Pi content in –Zn) were calculated and presented in Figure 6. These new results show a significant difference in lyso-PC/PC ratio between Col-0 and Sap-0 accessions in –Zn.

7) With regard to LPCAT1, a SNP explaining 11% of shoot Pi variation in -Zn was found. However, details on the contribution of the PHT1 loci to this trait are not provided. A more detailed description of the GWAS results must be provided for the latter (significant SNPs specific of -Zn in all PHT1s?).

A new Supplementary file 2 was added to report the genome coordinates of all significantly associated SNPs in that region.

8) How sure are the authors that there are only two causal alleles of LPCAT1 (Col-0/SAP). This seems to be an implicit claim but rarely are causal genes dimorphic in Arabidopsis and more often they have multiple alleles. This does not take away from the causation they have found but they should definitely be clear that there may be other alleles in the text or they could conduct a phylogenetic analysis of the promoter and ORF to classify their alleles within a broader context.

This is an excellent suggestion. We performed a haplotype analysis which revealed that there is one major haplotype associated with the SNP that is associated with higher Pi upon -Zn. We have therefore added a new supplementary figure (Figure 1—figure supplement 3) and added this to the Results section.

9) In several cases the text requires modification to improve accuracy. Please modify the text to address the following points.a) The authors should be a bit careful in their claims. For example, in the subsection “GWAS identify two candidate genes involved in the accumulation of Pi in the shoot under Zn deficiency”, this RT-PCR shows that Col-0 is responding to the Zn limitation but doesn't inherently mean that all accessions are responding in the same way or even at all to the Zn limitation. It is a nice control but is only a control for Col-0.

The sentence was rewritten as follows: “As expected, Zn deficiency in shoots of Col-0 plants was associated with the induction of the expression of two Zn-deficiency marker genes, ZIP4 and ZIP12 (Jain et al., 2013) (Figure 1—figure supplement 1)”.

b) Similarly, in the subsection “GWAS identify two candidate genes involved in the accumulation of Pi in the shoot under Zn deficiency”, yes the heritability is high but the fact that there is a difference in the presence or absence of Zn means that this is really a G X E trait and not largely genetic. To make the largely genetic claim would require a G + E + G x E analysis to partition the variance.

The sentence was rewritten as follows: “The broad-sense heritability (H^2^) of the shoot Pi concentrations was 0.63 and 0.47 under +Zn and –Zn conditions, respectively”.

c) In the subsection “GWAS identify two candidate genes involved in the accumulation of Pi in the shoot under Zn deficiency”, SNP effect estimates are notoriously difficult to make accurate so I would suggest caution in using 11% vs. 49% to say this gene is the main effect. The Manhattan plots suggest that there are likely a number of other loci and if there is trans-LD between these loci as found by Brachi and Bergelson for defense metabolism, then individual SNP estimates are inflated as they capture the causation at other SNPs. This in no way diminishes the fact that they have found a gene involved. It is just better to be accurate in claims than to inflate claims.

The sentence was omitted.

d) Sometimes, the logic of the experimental decisions was hidden in the text. For example, in the first paragraph of the subsection “LPCAT1 acts downstream of bZIP23 transcription factor”, the authors find a ZDRE in LPCAT1 which helps to support the GWA association. Yet this section is introduced by saying that the goal was to understand the regulatory context which is a very general goal that implies the section will discuss all the promoter elements. It seems that if the focus is solely on ZDRE, that it would make sense to state that the goal was to provide more molecular causational links between Zn/LPCAT/Pi which hasn't really been well understood.

We have rewritten the sentence as follows: “To investigate the molecular causational links between Zn/LPCAT1/Pi we analyzed the *cis*-regulatory elements present within the 1500-bp region upstream of the *LPCAT1* start codon (in Col-0 background) using the search tool AthaMap (Bülow et al., 2010).”

10) Some important details are missing from the data and some figures are incorrect and/or the presentation is poor. Please address these points:a) The coordinates for the significant SNP associated with Pi concentration in shoots in Zn- conditions must be provided.

A new Supplementary file 2 was added to report the genome coordinates of all SNPs.

b) The two motifs (shown in Figure 2) are on the opposite strand from the ATG, so that figure is incorrect. The figure should be corrected.

The figure was corrected (same as point 13).

c) Figure 1. Please plot the data rather than showing histograms.

We plotted the data: Pi (+Zn) versus Pi (-Zn) in Figure 1. The histograms were kept as Figure 1—figure supplement 2.

d) Figure 6. The legend indicates that expression in both high and low Zn is shown but this is not the case. Please show the relative expression in both conditions.

The mistake is now corrected and the relative expression of *PHTs* in plant grown in low and high Zn presented is Figure 7 and Figure 7—figure supplementary 5 respectively.

e) An additional supplementary table listing the sequence of the newly identified ZDRE in all accessions analyzed should be included to make correlations with their shoot Pi content in +Zn/-Zn.

We included Supplementary file 4 listing the new ZDRE motif in the accessions analyzed, with the shoot Pi content in both conditions (+Zn and –Zn).

11) Please consider the following comment which may help strengthen the PHT analyses:For the PHT analyses, it seems odd to rely on a Bonferroni when the authors have direct evidence that this is likely too conservative and causing false negative results as shown from the PHT1 data. There are plenty of papers showing that Bonferroni is overly conservative. I would suggest using a more realistic p-value and then the authors could bring in the PHT transporters as a part of the GWA causal network. Right now it has an artificial spurious argument of saying "they weren't significant but almost were". An option would be to use the modified Bonferroni approach where you estimate the true number of independent SNPs within a dataset and use that as the denominator rather than the total number of SNPs which is really inflates the correction.

This is a good point. We now used a non-conservative FDR threshold of 10% that includes the *PHT* genes.

12) It is surprising that you have not used 1001 Genomes imputed SNPs for your GWAS. Please rerun the analysis and report whether the apparently causal SNP is among the best hits. Consultation of the 1001 Genomes website indicates that this SNP is well represented in the 1001 Genomes data, with ~80% accessions having the standard allele, ~10% the alternative allele, and ~10% missing data. You should also show directly the Pi data for the accessions you analyzed, comparing distribution of Pi levels in all of your accessions with the standard versus all accessions with the alternative allele.

We started this project and conducted the GWASs before the 1001 genome paper was published. Based on the reviewer’s request, we have generated Supplementary file 3 which clearly shows that the 1001 genome data is consistent with the causality of our SNP in the motif. We have also further used the 1001 genomes sequence data to perform GWAS on the full set of accessions, but there was no SNP reaching genome wide significance (FDR < 10%). However, due to population structure correction it can be expected that there is a notable amount of false negatives. Given the experimental evidence that we provide, we don’t think that an absence of this peak in the 1001 genome based GWAS detracts from our conclusions. Finally, as recommended by the reviewer, we have provided the Pi data for *Arabidopsis thaliana* accessions used in our study (Supplementary file 1). We included Supplementary file 3 listing the new ZDRE motif in the accessions analyzed, with the shoot Pi content in both conditions (+Zn and –Zn). Data contained in Supplementary file 3 are illustrated in a new Figure 3—figure supplement 4. Taken together, our data provide a compelling evidence that sequence variation in the motif is associated with Pi concentrations in shoots of Arabidopsis grown in different Zn conditions.

13) Please give the genome coordinates for all sequences discussed in the paper, especially the putatively causal SNP. Note also that Figure 2 has a significant error, as the ATG and the two cis motifs are on opposite strands. Please correct. In addition, please discuss that the apparently causal motif is immediately upstream of the ATG and in the 5' UTR. Briefly discuss what is known about transcription factors binding in the 5' UTR and acting as cis-regulators.

Two new Supplementary file 2 was added to report the genome coordinates of all SNPs.

The Figure 2 was corrected.

We included a new paragraph on the available knowledge about the importance of 5’UTR on the regulation of gene expression and about transcription factors binding the 5’UTR was added in the Discussion section: “Based on our results we propose that sequence change in 5’UTR sequence impacts gene(s) expression, and consequently causes variation in the gene(s) associated traits. […] Taken together, our study has uncovered a new example for the role of 5’UTR in gene expression regulation and evolution in plants.»